# Evolutionary analysis of swimming speed in early vertebrates challenges the 'New Head Hypothesis'

Humberto G. Ferrón [1,2✉] & Philip C. J. Donoghue [1✉]

The ecological context of early vertebrate evolution is envisaged as a long-term trend towards increasingly active food acquisition and enhanced locomotory capabilities culminating in the emergence of jawed vertebrates. However, support for this hypothesis has been anecdotal and drawn almost exclusively from the ecology of living taxa, despite knowledge of extinct phylogenetic intermediates that can inform our understanding of this formative episode. Here we analyse the evolution of swimming speed in early vertebrates based on caudal fin morphology using ancestral state reconstruction and evolutionary model fitting. We predict the lowest and highest ancestral swimming speeds in jawed vertebrates and microsquamous jawless vertebrates, respectively, and find complex patterns of swimming speed evolution with no support for a trend towards more active lifestyles in the lineage leading to jawed groups. Our results challenge the hypothesis of an escalation of Palaeozoic marine ecosystems and shed light into the factors that determined the disparate palaeobiogeographic patterns of microsquamous versus macrosquamous armoured Palaeozoic jawless vertebrates. Ultimately, our results offer a new enriched perspective on the ecological context that underpinned the assembly of vertebrate and gnathostome body plans, supporting a more complex scenario characterized by diverse evolutionary locomotory capabilities reflecting their equally diverse ecologies.

[1] Palaeobiology Research Group, School of Earth Sciences, University of Bristol, Life Sciences Building, Tyndall Avenue, Bristol BS8 1TQ, UK. [2] Cavanilles Institute for Biodiversity and Evolutionary Biology, University of Valencia, Paterna, 46980 Valencia, Spain. ✉email: humberto.ferron@bristol.ac.uk; phil.donoghue@bristol.ac.uk

The origin and early evolution of vertebrates is associated with a fundamental embryological revolution and recurrent rounds of whole genome duplication[1–4]. In this context, the 'New Head Hypothesis'[5–10] proposes that the emergence of neurogenic placodes, as well as neural crest and its novel derivative cell fates, brought about a remodelling of the head of invertebrate chordates and the development of novel sensory and anatomical structures, underpinning early vertebrate evolution. The ecological scenario underlying this evolutionary hypothesis is envisioned as a long-term trend from suspension feeding, in a *Branchiostoma*-like proto-vertebrate, towards increasingly active and predatory forms[5–7,11], culminating in the emergence of the first jawed groups (i.e., placoderms)[12,13] and the subsequent diversification of the crown-gnathostome clades that dominate current vertebrate diversity (i.e., osteichthyans and chondrichthyans)[14,15]. Most of the changes leading to the origin of jawed vertebrates are interpreted as adaptations for improved ventilation and locomotory capabilities[16–19], acquired successively, as evidenced by the many phylogenetic and anatomical intermediate groups of 'ostracoderms' (i.e., jawless stem-gnathostome lineages)[1] (Fig. 1). However, ostracoderm biology is poorly constrained[20] and inferences of the ecological mode of ostracoderms, on which the New Head ecological scenario was constructed, are largely anecdotal[10] or, at best, contested[21–29]. This occurs because many aspects of ostracoderm anatomy lack modern analogues. An exception to this is caudal fin architecture which has been shown to accurately predict swimming speed in living jawed fishes with independence of other potential contributing factors[30–32]. Here, we extend this approach to living jawless fishes which have a distinct caudal fin structure from jawed fishes, like that of many ostracoderm stem-gnathostomes,

demonstrating the same predictive value. Within this Extant Phylogenetic Bracket[33] we apply this approach to calculating swimming speeds in the ostracoderms, as a means of testing the New Head scenario of early vertebrate evolution as being characterized by an ecological trend toward increasingly active food acquisition[5–10].

To achieve this, we present a new modeling approach for predicting swimming speeds in living fishes based on a number of caudal fin metrics, implemented using phylogenetic generalized linear models. Previous models are limited in their applicability as they are based in datasets with low taxonomic diversity and/or do not account for phylogeny[30,31]. We use this framework to explore the phylogenetic and temporal patterns that characterize the evolution of swimming speed in early vertebrate evolution, as a key component and predictor of their locomotory capabilities, by applying ancestral character state reconstruction and evolutionary model fitting.

## Results

### Building a model with living taxa to predict swimming speeds.
We compiled 160 swimming speed records for 61 living fish species, including both osteichthyans and chondrichthyans with body lengths ranging from 4 cm to 105 cm and representing a wide range of ecologies, swimming modes and locomotion types (Table S1). Using this dataset, we performed different phylogenetically informed regressions (PGLS) for predicting swimming speeds by checking combinations of multiple predictors (i.e., total body length, caudal fin shape metrics, swimming conditions and several locomotion parameters). Model selection analysis based on Akaike Information Criterion (AIC) shows that swimming

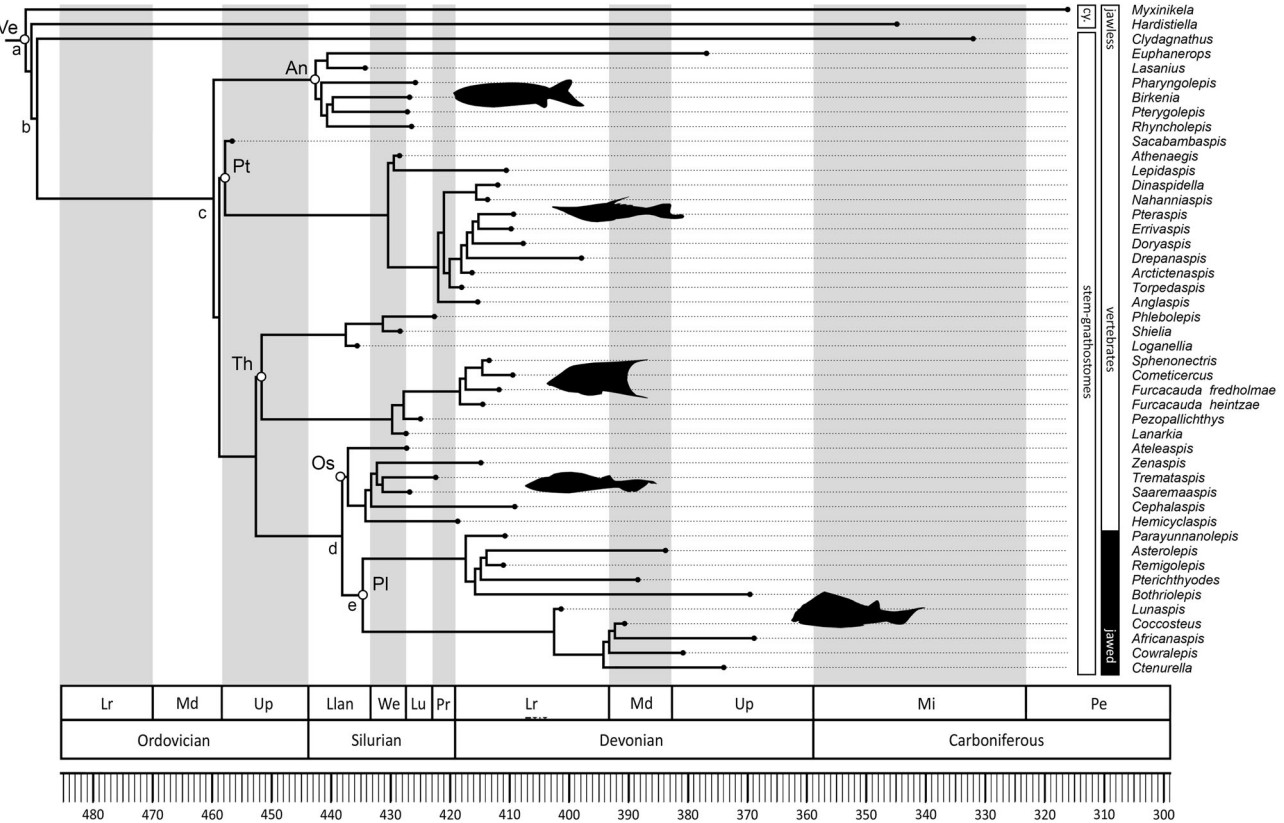

**Fig. 1 Time-calibrated phylogenetic tree including all the Palaeozoic early vertebrate taxa considered in the present study.** Taxa: Ve Vertebrata; An Anaspida; Pt Pteraspidomorphi; Th Thelodonti; Os Osteostraci; Pl Placodermi. Characters: a unmineralized skeleton; b mineralized dermal skeleton; c ventro-lateral fins; d perichondral bone, mineralized endoskeleton, pectoral fins and girdles, epicercal tail and cellular bone; e jaws, 'teeth', pelvic fins, and girdles. Timescale: Lr Lower; Md Middle; Up Upper; Llan Llandovery; We Wenlock; Lu Ludlow; Pr Pridoli; Mi Mississippian; Pe Pennsylvanian.

**Table 1 Fitting of phylogenetically informed regressions.**

| | $R^2$ | AIC | ΔAIC | wAIC |
|---|---|---|---|---|
| Speed ~ Length | 0.22 | 183.63 | 183.14 | 7.66E-41 |
| Speed ~ Length + Mode | 0.73 | 20.91 | 20.42 | 1.65E-05 |
| Speed ~ Length + Mode + LocType | 0.75 | 11.42 | 10.94 | 1.89E-03 |
| Speed ~ Length + Mode + AR | 0.75 | 6.06 | 5.57 | 2.77E-02 |
| Speed ~ Length + Mode + CircCF | 0.74 | 13.18 | 12.69 | 7.88E-04 |
| Speed ~ Length + Mode + RoundCF | 0.73 | 22.79 | 22.30 | 6.44E-06 |
| Speed ~ Length + Mode + SolCF | 0.75 | 10.89 | 10.41 | 2.47E-03 |
| Speed ~ Length + Mode + HeWiCF | 0.76 | 4.53 | 4.04 | 5.95E-02 |
| Speed ~ Length + Mode + HeWiCF + SolCF + RoundCF + CircCF + AR | 0.76 | 8.58 | 8.09 | 7.86E-03 |
| Speed ~ Length + Mode + HeWiCF + AR | 0.76 | 6.45 | 5.97 | 2.27E-02 |
| Speed ~ Length + Mode + LocType + AR | 0.76 | 11.38 | 10.89 | 1.94E-03 |
| Speed ~ Length + Mode + LocType + CircCF | 0.75 | 12.99 | 12.50 | 8.66E-04 |
| Speed ~ Length + Mode + LocType + RoundCF | 0.76 | 12.82 | 12.34 | 9.40E-04 |
| Speed ~ Length + Mode + LocType + SolCF | 0.75 | 13.31 | 12.82 | 7.36E-04 |
| Speed ~ Length + Mode + LocType + HeWiCF | 0.76 | 9.56 | 9.08 | 4.80E-03 |
| Speed ~ Length + Mode + LocType + HeWiCF + SolCF + RoundCF + CircCF + AR | 0.77 | 14.20 | 13.72 | 4.71E-04 |
| Speed ~ Length + Mode + LocType + HeWiCF + AR | 0.76 | 11.56 | 11.07 | 1.77E-03 |
| Speed ~ Length + Mode + HeWiCF + Group | 0.76 | 4.44 | 3.95 | 6.22E-02 |
| **Speed ~ Length + Mode + HeWiCF + Cond** | **0.77** | **0.49** | **0.00** | **4.49E-01** |
| Speed ~ Length + Mode + HeWiCF + Group + Cond | 0.77 | 1.51 | 1.02 | 2.69E-01 |
| Speed ~ Length + Mode + LocType + HeWiCF + Group | 0.76 | 8.13 | 7.64 | 9.85E-03 |
| Speed ~ Length + Mode + LocType + HeWiCF + Cond | 0.77 | 5.72 | 5.23 | 3.28E-02 |
| Speed ~ Length + Mode + LocType + HeWiCF + Group + Cond | 0.77 | 5.20 | 4.71 | 4.25E-02 |

The best model is shown in bold. wAIC Akaike weight. Predictors: *Length*, total body length; *Mode* swimming mode (burst, cruising); *LocType* locomotion type (anguilliform, carangiform, median/paired fin propulsion, thunniform); *AR* caudal fin aspect ratio, *CircCF* caudal fin circularity, *RoundCF* caudal fin roundness, *SolCF* caudal fin solidity, *HeWiCF* caudal fin height to width ratio, *Cond* swimming conditions (free swimming, non-free swimming).

speed is best explained by body length, swimming mode (burst, cruising), caudal fin height to width ratio, and swimming conditions (free, non-free-swimming) ($R^2 = 0.77$, Akaike weight = 0.45) (Tables 1 and S1, Fig. S1). Cross-validation analyses support that the best fitted model is robust and has a high predictive power ($R^2 = 0.73$) (Table S1 and Fig. S2). In addition, predictions performed on an independent dataset of living lampreys closely match records of swimming speeds reported in the literature ($R^2 = 0.97$) (Fig. S3 and Table S2). This validates the application of this model in fishes not only with epicercal tails, but also hypocercal tails (i.e., with caudal fins where the dorsal lobe is larger or smaller than the ventral one, respectively), such as in many (but not all) of ostracoderm groups[34].

**Reconstructing ancestral swimming speeds in early vertebrates.** From this PGLS model, we next inferred the cruising swimming speed of 41 early vertebrate taxa with well-known postcranial anatomy, including Palaeozoic cyclostomes (Myxinidae and Petromyzontidae), jawless stem gnathostomes (Conodonta, Anaspida, Pteraspidomorphi, Thelodonti and Osteostraci), and a representative sample of jawed stem gnathostomes (Placodermi) (Fig. 1 and Table S3). Cruising swimming speed was considered here as it represents a good approximation of activity and metabolic level in living taxa[35–39].

We performed ancestral character state reconstruction of cruising swimming speeds in a sample of 4500 trees accounting for both phylogenetic and temporal uncertainty (i.e., considering alternative phylogenetic hypotheses and fossil calibrations) (Fig. 2a). Swimming speeds inferred for the ancestral nodes of the main clades varied considerably, with the highest values predicted for anaspids (mean ± SD equal to 1.65 ± 0.06 body lengths per second, BL/s) and the lowest predicted for osteostracans, placoderms, and pteraspidomorphs (mean ± SD equal to 1.17 ± 0.03 BL/s, 1.15 ± 0.03 BL/s and 1.13 ± 0.10 BL/s, respectively). Intermediate speed values are predicted for thelodonts (1.30 ± 0.07 BL/s) which largely overlap with the

cruising swimming speeds predicted for the ancestral node of all vertebrates (mean ± SD equal to 1.29 ± 0.06 BL/s). In the following, we examined whether differences in body size impact on our estimates. When normalizing results by body length (i.e., considering all taxa being 0.1 meter in total body length) the pattern remains approximately the same, with the only exception being the ancestral node of all vertebrates that shows comparatively lower speeds (means ± SD equal to 1.65 ± 0.03, 1.45 ± 0.02, 1.43 ± 0.03, 1.41 ± 0.01, 1.39 ± 0.01, and 1.38 ± 0.02 BL/s for anaspids, thelodonts, pteraspidomorphs, osteostracans, placoderms, and vertebrates, respectively) (Fig. 2b).

We explored whether inferred ancestral morphologies would impact on ancestral estimates of swimming speed. For this, ancestral cruising swimming speeds of the main clades were also inferred by reconstructing the caudal fin morphology of their respective ancestral nodes, in 1000 trees randomly sampled from the original pool, using geometric morphometrics and posteriorly including those reconstructions into the established PGLS model (Fig. 2c). The results from this approach are remarkably similar to those derived from direct ancestral reconstruction of swimming speeds. Thus, anaspids show the highest ancestral cruising swimming speed values, followed by thelodonts, pteraspidomorphs, vertebrates, osteostracans, and placoderms, in decreasing order of magnitude (means ± SD equal to 1.453 ± 0.016, 1.360 ± 0.010, 1.331 ± 0.005, 1.332 ± 0.010, 1.317 ± 0.007, 1.290 ± 0.004 and 1.281 ± 0.005 BL/s, respectively).

This general pattern is also manifest when ancestral reconstruction analysis is repeated in all the original source trees, representing the main previously proposed hypotheses on the interrelationships of these taxa and accounting for alternative topologies potentially not captured in our pool of trees (Fig. S4).

**Fitting models of evolution to identify potential trends in swimming speed capabilities.** We fitted and compared two different evolutionary models to the cruising swimming speed datasets of early vertebrates using the pool of 4500 trees (i.e., one

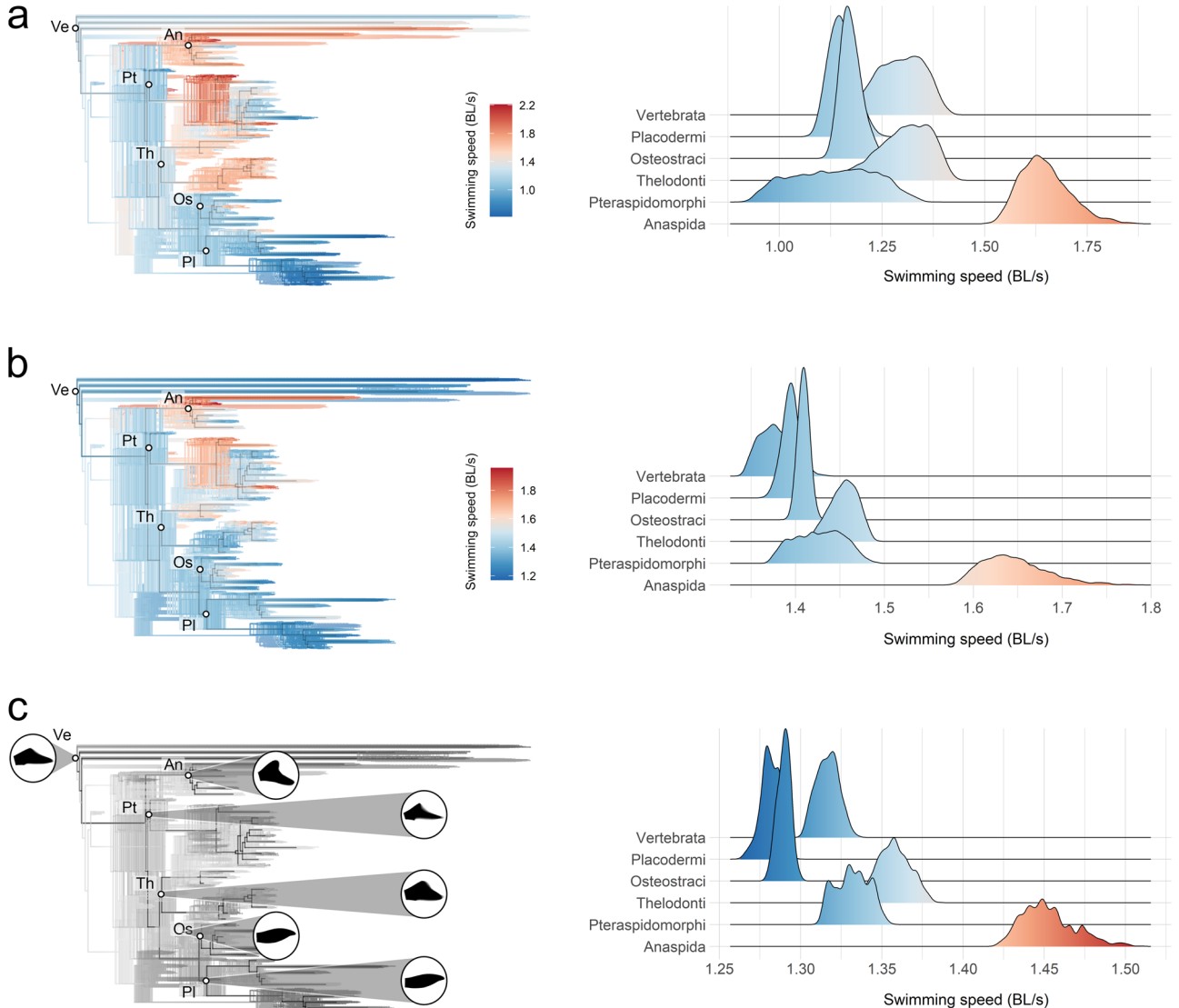

**Fig. 2 Ancestral cruising swimming speeds of Palaeozoic early vertebrates.** Results derived from ancestral character state reconstruction of (**a**) non-size-normalized speeds, (**b**) size-normalized speeds, and (**c**) caudal fin morphology. The outcomes of each analysis are summarized both as density trees with mapped ancestral speeds (in **a** and **b**) and caudal fin morphologies (in **c**); and density plots showing the ancestral speeds predicted for the main clades (left and right panels, respectively). Density trees include a subsample of 100 trees randomly selected from the original pool, while density plots are based on the whole pool of trees. Black outlines in (**c**) represent the average of all the caudal fin morphologies inferred for each selected node. Swimming speeds are in body lengths per second (BL/s). Taxa: Ve Vertebrata; An Anaspida; Pt Pteraspidomorphi; Th Thelodonti; Os Osteostraci; Pl Placodermi.

model including a directional drift or 'trend' component and the other one considering evolution under Brownian motion). When considering non-size-normalized cruising swimming speeds (Fig. 3a), rate of evolution ($\sigma^2$) parameter estimates are very similar for both models ($\sigma^2_{BM} = 0.0116 \pm 0.0010$; $\sigma^2_{Drift} = 0.0115 \pm 0.0010$) while slightly lower mean parameter values are estimated for the Brownian motion model ($\theta_{BM} = 1.2987 \pm 0.0500$; $\theta_{Drift} = 1.4493 \pm 0.2191$). The drift parameter estimated for the drift model is, on average, negative but very close to zero ($-0.0016 \pm 0.0019$). AIC scores suggest that Brownian motion model fits the data better than the drift model for all considered phylogenetic trees (note in Fig. 3a that AIC $_{BM}$ is usually smaller than the AIC $_{Drift}$ and, when larger, the difference is less than two units; $\Delta AIC = -1.7488 \pm 0.4157$). When considering size-normalized cruising swimming speeds, the pattern remains mostly the same, with a more marked difference on the mean parameter values estimated for each model ($\sigma^2_{BM} = 0.0027 \pm 0.0003$; $\sigma^2_{Drift} = 0.0027 \pm 0.0003$, $\theta$

$_{BM} = 1.3750 \pm 0.0185$; $\theta_{Drift} = 1.4817 \pm 0.024$; $\theta$ drift $_{Drift} = -0.0012 \pm 0.0003$; $\Delta AIC = -0.6877 \pm 0.1703$) (Fig. 3b).

## Discussion

Our methodological framework allows us to constrain for the first time the relationship between swimming speed, body size and caudal fin morphology in living non-tetrapod vertebrates accounting for their phylogenetic relationships (Table 1). The best fitting model shows a high predictive power of precisely inferring swimming speed, when considering both the original dataset and the leave-one-out cross-validation procedure (Figs. S1 and S2 and Table S1), outperforming most of the previous approaches for predicting swimming speeds in fishes[30,31,36,40]. As such, it lays the foundations for future studies aiming to assess locomotion aspects from morphological data in both living and extinct taxa and represents a valuable tool for interrogating long-standing discussions on the ecological scenarios that attempt to explain early vertebrate evolution.

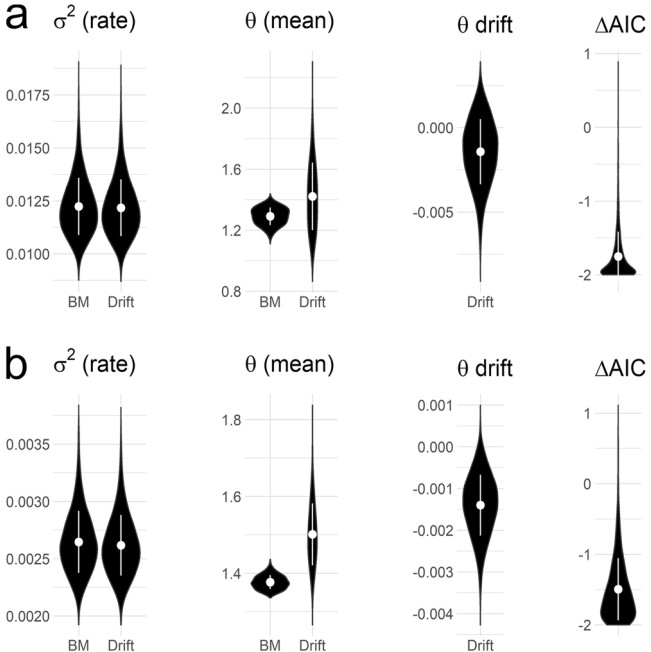

**Fig. 3 Evolutionary model fitting of cruising swimming speed in Palaeozoic early vertebrates.** Parameter estimates ($\sigma^2$, rate of evolution; $\theta$, trait mean; and $\theta$ drift, drift of the trait mean) for Brownian motion and drift evolutionary models fitted to the datasets of (**a**) non-size-normalized speeds and (**b**) size-normalized speeds, respectively. $\Delta$AIC represents the difference between AIC $_{BM}$ and AIC $_{Drift}$.

By applying this model to extinct Palaeozoic cyclostomes and stem gnathostomes, we reveal that the early evolution of vertebrates was characterized by diverging locomotory capabilities (Fig. 2). All our analyses predict that thelodonts and anaspids ancestrally had the highest swimming speed capabilities among all other major groups of stem-gnathostomes considered here, whereas the first jawed vertebrates (i.e., placoderms) had the lowest (or one of the lowest) swimming speeds. These results are supported by two alternative approaches (i.e., by direct ancestral reconstruction from the swimming speeds predicted in the tips and by predicting swimming speeds in ancestrally reconstructed caudal fin morphologies) and remain unaltered when accounting for the potential effect of body size (Fig. 2), as bigger fishes tend to have lower relative swimming speeds than the smaller fishes[41]. When comparing the fit of different evolutionary models to our data, those assuming no trend (i.e., Brownian motion) explain better the trait distribution, given a pool of trees accounting for both phylogenetic and temporal uncertainty. A model considering a subtle drift towards decreasing swimming speed has the lowest AIC values in only a few instances. However, BM is still equally supported given that most of the $\Delta$AIC values remail far below two units (Fig. 3).

Our study shows that lineages of microsquamous taxa (i.e., thelodonts and anaspids) had higher swimming capabilities than those characterized by the presence of rigid bony carapaces (macrosquamous ostracoderms e.g., pteraspidomorphs, osteostracans, and placoderms) (Figs. 2 and S4). This provides robust support for previous proposals of more active lifestyles in thelodonts and anaspids[25,26], with higher potential for migration, dispersal and colonization of a wider range of habitats[26,42]. Therefore, this finding offers a causal framework for understanding the contrasting palaeobioegeographic patterns that characterize each of these groups, where heavily armoured lineages have distributions marked by a strong endemism limited to areas connected by shallow continental shelfs, whereas microsquamous taxa had more cosmopolitan distributions resulting from the crossing of deep-water oceanic basins[43-48].

The prediction of high swimming capabilities as the ancestral condition for anaspids and thelodonts demonstrates that the rise of active nektonic vertebrates long-predated the Devonian[49], challenging the hypothesis of an escalation of Palaeozoic marine ecosystems[50]. In consequence, our results reinforce the view that the colonization of the pelagic realm by vertebrates might have follow more complex patterns than previously believed[51]. Ultimately, our results allow us to reject the hypothesis of a long-term evolutionary trend towards higher swimming capabilities in stem-gnathostomes, a pattern expected from the prevailing view that early vertebrate evolution was characterized by a stepwise acquisition of increasingly active lifestyles and modes of food acquisition[5-7,11,16-19]. In detail, however, our data reveal a more complex pattern in which the assembly of vertebrate and gnathostome body plans was characterized by diverse evolutionary trajectories for locomotory capabilities, ecologies[25-28], and feeding mechanisms[21-24,29].

## Method

**Dataset of living taxa**. Our dataset consists of 160 swimming speed records of 61 living osteichthyan and chondrichthyan species compiled from the literature (Table S1). For each record, we collected information relative to the fish total body length, swimming conditions (free, non-free swimming with specimens under controlled experimental conditions), locomotion type (anguilliform, carangiform, thunniform, median/paired fin locomotion[52]), swimming mode (cruising, burst), and different caudal fin morphological metrics, including the height to width ratio, circularity [$4*\pi*$(area/perimeter^2)], roundness [$4*$area/($\pi*$major_axis^2)], solidity [area/convex area], and aspect ratio [(height^2)/area]. When only standard or fork length measurements were available, those were transformed to total body lengths by implementing species-specific equations (Table S1). Caudal fin morphological variables were measured in ImageJ v 1.53b[53] by creating a macro (i.e., recording a series of commands using the command recorder) automating batch-processing of binary caudal fin outlines obtained from photographs of fishes in lateral aspect. Species-specific body length equations, photographs, and information regarding the locomotion type of each species were obtained from FishBase[54].

**Phylogenetic generalized least squares models**. Several phylogenetically informed regressions (PGLS) were performed for predicting swimming speeds by checking combinations of multiple predictors (i.e., total body length, caudal fin shape metrics, swimming conditions and several locomotion parameters). Akaike

information criterion (AIC) was employed to compare the goodness of fit for the different models. PGLS analyses were implemented in the package 'caper'[55] using R[56]. Multicollinearity was checked by calculating variance-inflation factors (VIF) and generalized variance-inflation factors (GVIF) with the R package 'car' v.3.0.12[57] and considering a threshold of $GVIF^{(1/(2*Df))} = 2$ (equivalent to VIF = 4)[58] (Table S4). The phylogeny employed in the PGLS analyses was built from previously published trees[59,60] by pruning taxa not included in our dataset using the R package 'ape'[61]. Species with multiple swimming speed records were included as polytomies that were resolved randomly with zero branch length. The predictive power of the best-fitting model was assessed by performing leave-one-out cross validation.

Our training dataset is composed of living taxa, the majority of which have symmetrical or epicercal caudal fins. Given that some pteraspidomorphs, anaspids and thelodonts possess hypocercal caudal fins, we explicitly evaluated the potential of our PGLS model for predicting swimming speeds in living fishes with this condition. With this in mind, we compared previously published records of swimming speeds in living lampreys (Table S2) with predictions derived from our model. The test sample included six records of three different species (*Entosphenus tridentatus*, *Petromyzon marinus* and *Lampetra fluviatilis*) and a range of body sizes comparable to that of the considered stem-gnathostome taxa.

**Dataset of extinct taxa and supertree construction**. A total of 41 early vertebrates, representing all taxa with well-known postcranial anatomy, were considered in our analyses. The dataset includes Palaeozoic cyclostomes (Myxinidae and Petromyzontidae), jawless stem gnathostomes (Conodonta, Anaspida, Pteraspidomorphi, Thelodonti and Osteostraci), and a representative sample of jawed stem gnathostomes (Placodermi) (Fig. 1 and Table S3).

A pool of 4500 phylogenetic supertrees including these taxa (plus Galeaspida) was generated accounting for both phylogenetic and temporal uncertainty. For this, we obtained 45 different topologies using matrix representation with parsimony in the R package 'phangorn'[62] from 37 source topologies, accounting for the main hypotheses on the interrelationships of these taxa present in the literature (Fig. S4). Each of the obtained trees was time calibrated 100 times using the R package 'paleotree'[63] by randomizing the tip age of every species within the chronostratigraphic unit, at age or subperiod rank, where their first appearance occurs. A minimum age constraint was set in the ancestral nodes of the main clades of stem-gnathostomes considering their first appearance in the fossil record (Conodonta, Furongian[64]; Anaspida, Llandovery[20]; Pteraspidomorphi, Darriwilian-Sandbian[65,66]; Thelodonti, Sandbian[26]; Osteostraci, Aeronian[67]; Placodermi, Telychian–Wenlock)[68].

**Ancestral cruising swimming speeds estimations**. The ancestral cruising swimming speeds of the main clades of early vertebrates were estimated following two alternative procedures. First, we performed ancestral character state reconstruction from cruising swimming speeds predicted for the tips. For this, caudal fin morphological variables were measured in the 41 fossil taxa from reconstructions in the literature (Table S3), following the same procedure than described above, and their cruising swimming speeds were inferred from the best fitted PGLS model. We considered free-swimming conditions and maximum body lengths reported in the literature (Table S3). Ancestral character state reconstruction analysis was performed in the pool of 4500 phylogenetic trees using maximum likelihood method implemented in the R package 'phytools'[69]. The same analysis was repeated with size-normalized cruising swimming speeds, considering all taxa having a total body length of 0.1 meters. Secondly, we reconstructed ancestral caudal fin morphologies and derived cruising swimming speed predictions from them. For this, we performed a geometric morphometric analysis on the caudal fin of the 41 early vertebrate taxa (excluding Galepasida where specimens with complete caudal fins are not known). We considered a total of 102 landmarks, including two landmark type I in the caudal fin base dorsal and ventral margins and 100 landmark type III equally interpolated along the caudal fin outline (Fig. S5), that were digitized using TpsDig v.2.26[70] on previously published reconstructions in lateral view (Table S3). Landmark coordinates of all specimens were fitted by implementing generalized Procrustes superimposition in the R package 'geomorph'[71] to remove variation in rotational, scale and translational differences between specimens. Ancestral morphologies where reconstructed in 1000 trees randomly selected from the original pool of trees also using 'geomorph'[71]. Caudal fin variables were then measured in the predicted morphologies and their cruising swimming speeds were inferred from the best fitted PGLS model. In these analyses, we considered free-swimming conditions and all taxa having a total body length of 0.1 meters. In all cases, results were visualized as density plots and density trees with mapped ancestral speeds using the R packages 'ggplot2'[72], 'ggridges'[73] and 'ggtree'[74]. We also visualized the average of all the ancestral caudal fin morphologies inferred for each node by obtaining Z projections in ImageJ v 1.53b[53].

Finally, we performed ancestral character state reconstruction from size-normalized cruising swimming speeds predicted for the tips in all 37 compiled source topologies using maximum likelihood method after time calibration in the R packages 'paleotree'[63] and 'phytools'[69].

**Fitting of evolutionary models**. We explored the presence of evolutionary trends in the swimming capabilities of early vertebrates by fitting an evolutionary model including a drift or 'trend' component to our datasets of both non-size-normalized and size-normalized cruising swimming. A second model considering Brownian motion (i.e., where the trait evolves via a 'random walk') was also fitted to the data and AIC was employed to compare the goodness of fit for both models. We estimated the rate of evolution ($\sigma^2$) and trait mean ($\theta$) by finding the maximum-likelihood parameter values for each model. The drift of the trait mean was also estimated for the drift model. The fit of evolutionary models and parameter estimations were carried out in the original pool of 4500 phylogenetic trees using the R package 'geiger'[75]. Results were presented as density plots with mean and standard deviation values using the R package 'ggplot2'[72].

**Statistics and reproducibility**. The results are expressed as mean ± standard deviation (SD) and $p < 0.05$ was considered significant.

**Reporting summary**. Further information on research design is available in the Nature Research Reporting Summary linked to this article.

## Data availability
Data sets (Supplementary Data 1) are available from the Figshare database. https://doi.org/10.6084/m9.figshare.16774747 [76].

## Code availability
R code is available from the Figshare database. https://doi.org/10.6084/m9.figshare.16774747 [76].

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

## Acknowledgements

We thank two anonymous reviewers for providing thoughtful and valuable comments on the manuscript. H.G.F. is recipient of a European Commission grant H2020-MSCA-IF-2018-839636. P.C.J.D. was funded by Natural Environment Research Council (NERC) grant (NE/P013678/1), part of the Biosphere Evolution, Transitions and Resilience (BETR) program, which is co-funded by the Natural Science Foundation of China (NSFC); Biotechnology and Biological Sciences Research Council (BBSRC) grant (BB/T012773/1); and Leverhulme Trust Research Fellowship (RF-2022-167).

## Author contributions

H.G.F. and P.C.J.D. conceived the project. H.G.F. analyzed the data and interpreted the results, and H.G.F. and P.C.J.D. wrote the manuscript.

## Competing interests

The authors declare no competing interests.
