## [Peer Review File · Communications Biology]

Reviewers' comments:

Reviewer #1 (Remarks to the Author):

Ferron and Donoghue challenge the implications of the 'New Head Hypothesis' particularly with respect to predictions of increasing swimming abilities and increasing predation through the vertebrates. They focus on caudal fin morphology, linking morphology to swimming speeds in various extant taxa that are then applied to fossil early vertebrates, both jawless and jawed (placoderms). The New Head Hypothesis would predict that among these, the placoderms would be fastest and most predatory, but this is not the case, instead, swimming and locomotory capabilities are much more complex among taxa such as the thelodonts, anaspids, heterostracans and osteostracans. Ferron and Donoghue and their colleagues are leading the way in new research on locomotion in largely neglected stem-gnathostomes, with this study focusing on the tail as opposed to previous studies investigating the skull/headfield.

I have some comments on the attached document, but my main comment is that placoderms are considered to be the slowest swimmers, but the sample size is small, with two antiarchs with long armour, and two arthrodires, one of which (*Cowralepis*) is flattened and possibly a specialized swimmer. Is this a broad enough sample to support your claims?

Reviewer #2 (Remarks to the Author):

In their manuscript entitled "Evolutionary analysis of swimming speed in early vertebrates challenges the "New Head Hypothesis" H.G. Ferron and P.C.J. Donoghue develop prediction models for swimming speed in an extant reference group based on body size, swimming mode, locomotion type and caudal fin shape. They use these models to infer swimming speeds for extinct groups of jawless vertebrates and several placoderm taxa and employ an ancestral state reconstruction approach to test whether the acquisition of certain anatomical features in the stem group of gnathostomes is linked to a increase in swimming speed - as assumed by proponents of the "New Head Hypothesis" - and whether the supposed rise of nektonic fish and active swimming in the Devonian was pre-dated by an earlier radiation of fast swimmers.

The study by Ferron and Donoghue represents an innovative, elaborate and (for the most part) well-documented approach whose results are well reproducible due to the data and script provided in the well-structured supplementary materials. The authors address an important research question relevant for a large group of readers including (but not restricted to) specialists for functional morphology, evolutionary biology, ichthyology, palaeoecology and, arguably, their manuscript is well suited for this journal. According to my perception, the state of knowledge concerning the "New Head Hypothesis" prior to their effort is quite well outlined (I am not a specialist for jawless fish, though).

While the study design should work in principle, a number of aspects makes the results difficult to interpret:

(1) Model for the inference of swimming speed: According to the coefficients of determination listed in Table 1, regression models including one or several morphological parameters for the caudal fin do not explain a higher percentage of variance in swimming speed than models which only consider body size, mode and locomotion type (in most cases $r^2 = 0.75 \pm 0.3$, even though some models appear to be somewhat better according to the AIC). - Why? Do I need to consider caudal fin morphometrics at all if I also have to make assumptions about the "mode" and "type" of swimming? Would the approach as a whole not be better if I could infer swimming speed of fossil taxa only based on body size and caudal fin morphology - without making assumption about "swimming mode" and "type" which are not directly observable in extinct groups? The authors should address the question whether/to what degree/why the presumably

independent variables for the swimming speed models are correlated and whether they can infer swimming speed without making assumptions about "mode" and "type".

(2) Sampling of early gnathostomes: This study considers only four relatively late occurring Devonian placoderms as representatives for the clade Gnathostomata. But all major groups of jawed fish, such as placoderms, acanthodians, elasmobranchs and osteichthyans, occur as early as Silurian and also the group Placodermi in the classical sense is not nearly sufficiently covered.* The long lineage connecting the placoderm ancestor and the last common ancestors of placoderms and osteostracans (nodes e and f) according to the tree depicted in Fig. 1 includes multiple nodes/ divergences of major groups and many potential changes in swimming speed. ...Sampling is also poor around the common ancestor of vertebrates - the included examples of fossil cyclostomes and conodonts have a Carboniferous to Permian age (tens to hundreds of millions of years after the first occurrence of vertebrates and after the divergence of gnathostomes and cyclostome groups) and I wonder whether the swimming capability in the earliest vertebrates could be better defined by including certain Cambrian chordates (Haikouichthys etc.) in this analysis.

*e.g. first occurrences/ranges of major clades according to the synopsis by Fernandez and Janvier (2021) on finned vertebrates:

https://www.researchgate.net/publication/357876245_Finned_Vertebrates

(3) Node ages. It is not entirely clear where the node ages - especially those used in Fig. 1 - come from and arguably a table of all major nodes/group divergence ages with references (e.g. citations for first occurrences/group ranges) should be added to the main text or to the supplementary material. Especially nodes d-f appear to be too young, e.g. node f, the last common ancestor of antiarchs and arthrodירים, should fall somewhere into the Silurian (ca. 435 Ma) and not late early Devonian (395 Ma).

Given the problems of sampling/lacking coverage of relevant groups (point 2) and mistaken node ages (point 3), the ancestral state reconstruction cannot provide an adequate picture of swimming speed changes on the lineage connecting the last common ancestors of vertebrates (node a) and gnathostomes, a problem which is exacerbated by many alternative phylogenies (as presented in the supplement).

In conflict with the authors' conclusion, a trend towards higher swimming capabilities cannot be declined (based on the data presented) for some parts of early vertebrate phylogeny, namely the clade of osteostracans and gnathostomes after the Llandovery. Especially if a certain temporal lag between the occurrence of a crucial innovation in the gnathostome stem group ("New Head" in the Silurian) and the realization of its benefits for locomotion within various groups of Devonian gnathostomes had occurred, such a trend in a "Devonian nekton revolution" could only be detected with a better coverage of Devonian gnathostomes.

Considering body size as a crucial factor in locomotion evolution, I wonder whether an important evolutionary novelty in stem gnathostomes and their gnathostome descendants could have led to a release of size constraints on swimming speed, enabling larger fish to become faster and/or fast fish to become larger. Thus, I would encourage the authors to consider body size in greater detail, e.g. in additional ASR as a separate character.

Despite these criticisms, I think that the authors have convincingly shown that some stem-gnathostome groups, such as thelodonts and anaspids, had higher swimming speeds than some (but perhaps not most) Devonian placoderms and that the occurrence of nektonic groups and capable swimmers did occur earlier than assumed according to the "New Head hypothesis". However, as of now, their ancestral state reconstruction approach is more harmful than helpful and I would recommend that the authors improve their database as outlined above (especially points 2 and 3) to put somewhat more weight behind their claims.

As an afterthought (I wonder whether I overlooked this in the supplementary material): In addition to listing the reconstructed values for swimming speeds at important nodes (ancestors of major groups, stem line nodes a-f) in a table as a part of the results section it is always helpful to account for the uncertainty of these reconstructed values by adding an error bar (e.g. through a bootstrapping approach which uses multiple values/distributions instead of individual values assigned to certain tree tips) - to illustrate whether supposed changes in a series of ancestors are really "notable" or within the margin of error (of the ASR approach).

Response to Referees

Reviewer #1 (Remarks to the Author):

“Ferron and Donoghue challenge the implications of the 'New Head Hypothesis' particularly with respect to predictions of increasing swimming abilities and increasing predation through the vertebrates. They focus on caudal fin morphology, linking morphology to swimming speeds in various extant taxa that are then applied to fossil early vertebrates, both jawless and jawed (placoderms). The New Head Hypothesis would predict that among these, the placoderms would be fastest and most predatory, but this is not the case, instead, swimming and locomotory capabilities are much more complex among taxa such as the thelodonts, anaspids, heterostracans and osteostracans. Ferron and Donoghue and their colleagues are leading the way in new research on locomotion in largely neglected stem-gnathostomes, with this study focusing on the tail as opposed to previous studies investigating the skull/headshield.

I have some comments on the attached document, but my main comment is that placoderms are considered to be the slowest swimmers, but the sample size is small, with two antiarchs with long armour, and two arthrodires, one of which (Cowralepis) is flattened and possibly a specialized swimmer. Is this a broad enough sample to support your claims?”

We acknowledge the referee's point and we have rerun all the analyses with a considerably larger sample of placoderms that cover the main clades with known caudal fin and the whole diversity of body and tail morphologies. This includes one Petalichthyda (*Lunaspis*), five Antiarchi (*Parayunnanolepis*, *Bothriolepis*, *Pterichthyodes*, *Remigolepis*, and *Asterolepis*), one Ptyctodontida (*Ctenurella*), one Phyllolepidida (*Cowralepis*), and two Arthrodira (*Cocosteus* and *Africanaspis*).

Please note that now we have included an important number of Antiarchi which constitute the earliest branching clade of jawed vertebrates. This is essential to properly constrain the nature of the last common ancestor (LCA) of all jawed vertebrates and therefore to test any trend expected from the New Head Hypothesis (i.e. in the line/branches between the LCA of all vertebrates and the LCA of all jawed vertebrates).

Different sections of the manuscript have been modified in order to accommodate these changes but note that the main conclusions of our study remain unaltered.

Line 102. 'Is this going to be biased with respect to preservation- ie, you're only sampling anatomies linked to certain environments, rather than more broadly?'

The existence of this bias in our dataset is very unlikely as the considered specimens come from a wide range of habitats, covering from continental to deep-water depositional environments (see data in Sallan et al. 2018).

Sallan, L., Friedman, M., Sansom, R. S., Bird, C. M., & Sansom, I. J. (2018). The nearshore cradle of early vertebrate diversification. *Science*, 362(6413), 460-464.

Line 109. 'Can you describe what these sources of uncertainty would be?'

Amended.

Lines 192-193. 'ecological scenarios that underlie'

Amended.

Line 199. 'But this is based on a small number of placoderms, 2 antiarchs and 2 arthodires, 3 of these with heavier armour, and 1 a flattened and presumably specialized placoderm with respect to locomotion. Can you use these taxa to generalize across placoderms? Would speeds in a group with reduced head and trunkshield armour like the ptyctodonts be faster?'

Amended, see comments above.

Line 213. 'I know that this would be beyond the scope of this work, but swimming capabilities are not going to be based only on the caudal fin, but also factors such as cranial shape, bone density and ornamentation, scale patterning and ornamentation etc. The authors are making substantial claims but is the whole story being considered?'

We agree with the reviewer that locomotory capabilities depends on multiple factors. However, caudal fin morphology by itself has been proven to be highly correlated with swimming speed, allowing to derive reliable estimations with independence of other variables (see references in the manuscript). This is now better clarified in the text (lines 50-52).

Line 222. 'We knew this already via acanthodians and the Chinese Silurian faunas including osteichthyans?'

We agree with the referee and have added a new reference to support that.

Reviewer #2 (Remarks to the Author):

In their manuscript entitled "Evolutionary analysis of swimming speed in early vertebrates challenges the "New Head Hypothesis" H.G. Ferron and P.C.J. Donoghue develop prediction models for swimming speed in an extant reference group based on body size, swimming mode, locomotion type and caudal fin shape. They use these models to infer swimming speeds for extinct groups of jawless vertebrates and several placoderm taxa and employ an ancestral state reconstruction approach to test whether the acquisition of certain anatomical features in the stem group of gnathostomes is linked to a increase in swimming speed - as assumed by proponents of the "New Head Hypothesis" - and whether the supposed rise of nektonic fish and active swimming in the Devonian was pre-dated by an earlier radiation of fast swimmers.

The study by Ferron and Donoghue represents an innovative, elaborate and (for the most part) well-documented approach whose results are well reproducible due to the data and script provided in the well-structured supplementary materials. The authors address an important research question relevant for a large group of readers including (but not restricted to) specialists for functional morphology, evolutionary biology, ichthyology, palaeoecology and, arguably, their manuscript is well suited for this journal. According to my perception, the state of knowledge concerning the "New Head Hypothesis" prior to their effort is quite well outlined (I am not a specialist for jawless fish, though).

While the study design should work in principle, a number of aspects makes the results difficult to interpret:

(1) Model for the inference of swimming speed: According to the coefficients of determination listed in Table 1, regression models including one or several morphological parameters for the caudal fin do not explain explain a higher percentage of variance in swimming speed than models which only consider body size, mode and locomotion type (in most cases $r^2 = 0.75 \pm 0.3$, even though some models appear to be somewhat better according to the AIC). - Why? Do I need to consider caudal fin morphometrics at all if I also have to make assumptions about the "mode" and "type" of swimming? Would the approach as a whole not be better if I could infer swimming speed of fossil taxa only based on body size and caudal fin morphology - without making assumption about "swimming mode" and "type" which are not directly observable in extinct groups? The authors should address the question whether/to what degree/why the presumably independent variables for the swimming speed models are correlated and whether they can infer swimming speed without making assumptions about "mode" and "type".

R^2 and AIC quantify different aspects of the model. Conceptually, the former reflects how well the model explains the observed data while the second explains how well the model will predict on new data. In any case, the selected model, which includes caudal fin metrics, has both the highest R^2 (although we agree it is only slightly higher than other models) and the lowest AIC ($\Delta AIC > 2$ respect to all other models, except for Speed ~ Length + Mode + HeWiCF + Group + Cond, and $\Delta AIC > 10$ respect to models that do not consider caudal fin metrics). The need of considering a model including caudal fin metrics is further evidenced when we run a likelihood ratio test between the best supported model and the same model excluding caudal fin measurements, where we find that the former is significantly superior:

Model 1: speed ~ length + mode + Exp

Model 2: speed ~ length + mode + HeWiCF + Exp

	Df	LogLik	Df	Chisq	Pr(>Chisq)
1	5	-4.3565			
2	6	5.7568	1	20.227	6.879e-06 ***

Signif. codes: 0 '***' 0.001 '**' 0.01 '*' 0.05 '.' 0.1 ' ' 1

The results of this analysis have not been included in the revised manuscript, but we are happy to do so if the editors and referee deemed it necessary.

On the other hand, all predictors in the model are directly observable and measurable in the fossils and no *a priori* assumptions are needed. The model with the highest support, used to derive predictions in fossils, includes body length, caudal fin metrics (HeWiCF), swimming mode (burst, cruising) and swimming conditions (free, non-free swimming) as predictors. The latter two predictors were included to account for any potential impact of these parameters in the swimming speeds of living taxa and their incorporation in the models is demonstrated to increase their predictive power. But note that when making predictions, the choice between cruising/burst and free/non free swimming is arbitrary and only depends on the swimming mode and conditions we are interested to infer.

Therefore, we have predicted swimming speeds in all fossil taxa under free-swimming conditions, as this would be the situation for Palaeozoic stem-gnathostomes, and cruising swimming, this being a good approximation of metabolic and activity level; but our model would also allow speed predictions for these taxa under non-free and/or burst swimming conditions. In any of these cases the general patterns provided here would remain the same given that the differences between swimming speed predictions

would remain proportional between taxa (resulting from their multiplication by the coefficients associated to each predictor). The reasons of our choice are now clarified in the text (lines 111-112).

Besides that, we have considered interesting to explore other parameters that could affect swimming speed but that are directly observable only in living taxa, such as locomotion type (anguilliform, carangiform, thunniform, median/paired fin locomotion). However, none of the models including those provided better fit, which ensures extrapolation of our methodological framework to make inferences in fossils with no need to assume non-directly measurable parameters on them.

Finally, regarding the reviewer's concern on the potential correlation among independent variables, we have checked multicollinearity using variation inflation factors. No important collinearity has been detected (i.e., $GVIF^{(1/(2 \cdot Df))} < 2$). These analyses and their results are now included in the revised version of the manuscript (lines 262-264) and in the new Supplementary Table 4.

Ferrón, H. G., Martínez-Pérez, C., & Botella, H. (2018). The evolution of gigantism in active marine predators. *Historical Biology*, 30(5), 712-716.

“(2) Sampling of early gnathostomes: This study considers only four relatively late occurring Devonian placoderms as representatives for the clade Gnathostomata. But all major groups of jawed fish, such as placoderms, acanthodians, elasmobranchs and osteichthyans, occur as early as Silurian and also the group Placodermi in the classical sense is not nearly sufficiently covered. The long lineage connecting the placoderm ancestor and the last common ancestors of placoderms and osteostracans (nodes e and f) according to the tree depicted in Fig. 1 includes multiple nodes/ divergences of major groups and many potential changes in swimming speed. ...Sampling is also poor around the common ancestor of vertebrates - the included examples of fossil cyclostomes and conodonts have a Carboniferous to Permian age (tens to hundreds of millions of years after the first occurrence of vertebrates and after the divergence of gnathostomes and cyclostome groups) and I wonder whether the swimming capability in the earliest vertebrates could be better defined by including certain Cambrian chordates (Haikouichthys etc.) in this analysis.*

**e.g. first occurrences/ranges of major clades according to the synopsis by Fernandez and Janvier (2021) on finned vertebrates:*

https://www.researchgate.net/publication/357876245_Finned_Vertebrates”

We are very grateful to the referee for raising these concerns. We have now implemented several changes to address these points. Accordingly, all the analyses have been rerun (1) constraining the age of the main nodes (according to their FADs) and (2) considering a substantially larger sample of placoderms that cover the main clades with known caudal fin and the whole diversity of body and tail morphologies. This includes one Petalichthyda (*Lunaspis*), five Antiarchi (*Parayunnanolepis*, *Bothriolepis*, *Pterichthyodes*, *Remigolepis*, and *Asterolepis*), one Ptyctodontida (*Ctenurella*), one Phyllolepidida (*Cowralepis*), and two Arthrodira (*Cocosteus* and *Africanaspis*). Placoderms are now generally accepted as a paraphyletic grade of jawed vertebrates and, as such, these samples represent successive sister lineages to crown-gnathostomes. As such, these samples allow us to infer the nature of a sequence of successive ancestors of the gnathostome crown-ancestor and descendants of the LCA of osteostracans and jawed vertebrates. As such, provide for a robust test of the New Head Hypotheses which was concerned with the transition from invertebrate chordates to jawed vertebrates.

Importantly, we now include a number of Antiarchi which constitute the earliest branching clade of jawed vertebrates. This is essential to properly constrain the nature of the last common ancestor (LCA)

of all jawed vertebrates and therefore to test any trend expected from the New Head Hypothesis (i.e. in the line/branches between the LCA of all vertebrates and the LCA of all jawed vertebrates). The condition of other major clades of crown-gnathostomes is beyond the scope of the New Head Hypothesis and, regardless, expected to have a negligible impact on the reconstruction of these nodes.

We have modified a number of different sections of the manuscript in order to accommodate these changes and have specified the ages and references consulted to constrain the node calibrations of the main clades of stem gnathostomes (lines 295-299). Note that the main conclusions of our study remain unaltered.

On the other, the caudal fin morphology of Cambrian vertebrates, such as *Myllokunmingia* and *Haikouichthys*, and Devonian cyclostomes, such as *Priscomyzon*, cannot be delimited given the presence of a continuous fin structure along the caudal and dorsal regions (Zhang and Hou, 2004; Gess et al. 2006). As consequence, these taxa cannot be included in our analyses. For this reason, we have also excluded from our dataset the lamprey *Mayomyzon* (Janvier 2008), which was considered in the previous version of the manuscript.

Bardack, D., & Zangerl, R. (1968). First fossil lamprey: a record from the Pennsylvanian of Illinois. *Science*, 162(3859), 1265-1267.

Gess, R. W., Coates, M. I., & Rubidge, B. S. (2006). A lamprey from the Devonian period of South Africa. *Nature*, 443(7114), 981-984.

Janvier, P. (2008). Early jawless vertebrates and cyclostome origins. *Zoological science*, 25(10), 1045-1056.

Zhang, X. G., & Hou, X. G. (2004). Evidence for a single median fin-fold and tail in the Lower Cambrian vertebrate, *Haikouichthys ercaicunensis*. *Journal of evolutionary biology*, 17(5), 1162-1166.

“(3) Node ages. It is not entirely clear where the node ages - especially those used in Fig. 1 - come from and arguably a table of all major nodes/group divergence ages with references (e.g. citations for first occurrences/group ranges) should be added to the main text or to the supplementary material. Especially nodes d-f appear to be too young, e.g. node f, the last common ancestor of antiarchs and arthrodירים, should fall somewhere into the Silurian (ca. 435 Ma) and not late early Devonian (395 Ma).”

Amended, see comments above.

“Given the problems of sampling/lacking coverage of relevant groups (point 2) and mistaken node ages (point 3), the ancestral state reconstruction cannot provide an adequate picture of swimming speed changes on the lineage connecting the last common ancestors of vertebrates (node a) and gnathostomes, a problem which is exacerbated by many alternative phylogenies (as presented in the supplement).”

Amended, see comments above. Our sampling includes almost all taxa that have known tail fins. Our results remain the same regardless of the hundreds of phylogenies considered. Thus, our conclusions are robust to phylogenetic uncertainty. Thus, to the contrary, the consideration of competing topologies is a strength of our analysis.

“In conflict with the authors' conclusion, a trend towards higher swimming capabilities cannot be declined

(based on the data presented) for some parts of early vertebrate phylogeny, namely the clade of osteostracans and gnathostomes after the Llandovery. Especially if a certain temporal lag between the occurrence of a crucial innovation in the gnathostome stem group ("New Head" in the Silurian) and the realization of its benefits for locomotion within various groups of Devonian gnathostomes had occurred, such a trend in a agreement with a "Devonian nekton revolution" could only be detected with a better coverage of Devonian gnathostomes."

No trend is supported by ancestral state character reconstruction and evolutionary model fitting after the inclusion of a larger sample of gnathostomes (see Figs. 2 and 3 and corresponding parts of the text).

"Considering body size as a crucial factor in locomotion evolution, I wonder whether an important evolutionary novelty in stem gnathostomes and their gnathostome descendants could have led to a release of size constraints on swimming speed, enabling larger fish to become faster and/or fast fish to become larger. Thus, I would encourage the authors to consider body size in greater detail, e.g. in additional ASR as a separate character."

We completely agree with the referee that size is a critical factor on relative swimming speed. Accordingly, this aspect was considered in a complementary analysis where we predicted size-normalized cruising swimming speeds that were used later for ancestral character state reconstructions and evolutionary model fitting (see Figs 2B and 3B and corresponding parts of the text).

We have focused on testing the existence of any trend between the nodes representing the LCA of all vertebrates and the LCA of all jawed vertebrates, as these delimit the branches of interest when designing specific tests for the New Head Hypothesis. We agree that the aspects raised by the referee are very interesting but exploring the evolutionary patterns of swimming speed and body size beyond the gnathostome crown ancestor is beyond the scope of the New Head Hypothesis and, therefore, our current study. However, we thank the referee for making these points which we will surely consider them in upcoming studies.

"Despite these criticisms, I think that the authors have convincingly shown that some stem-gnathostome groups, such as thelodonts and anaspids, had higher swimming speeds than some (but perhaps not most) Devonian placoderms and that the occurrence of nektonic groups and capable swimmers did occur earlier than assumed according to the "New Head hypothesis". However, as of now, their ancestral state reconstruction approach is more harmful than helpful and I would recommend that the authors improve their database as outlined above (especially points 2 and 3) to put somewhat more weight behind their claims."

Respectfully, we disagree. We have sampled the phylogenetic breadth of taxa encompassed by the New Head Hypothesis. We would have liked to have included more species but, unfortunately, caudal fin morphology is known for only a limited number of taxa (e.g. it is not yet known for any galeaspids). Therefore, we have included all available data. Our results and our interpretations of them are readily testable based on new discoveries of tail anatomy among stem-gnathostomes.

As an afterthought (I wonder whether I overlooked this in the supplementary material): In addition to listing the reconstructed values for swimming speeds at important nodes (ancestors of major groups, stem line nodes a-f) in a table as a part of the results section it is always helpful to account for the uncertainty of these reconstructed values by adding an error bar (e.g. through a bootstrapping approach which uses multiple values/distributions instead of individual values assigned to certain tree tips) - to illustrate whether supposed changes in a series of ancestors are really "notable" or within the margin of error (of the ASR approach)."

Amended, this information is now included in the online repository together with the R code (<https://figshare.com/s/cefdb9a21f0804ef6a37>).

We are extremely grateful to the referees for their time and insight. We hope that these edits have resolved any issues with the manuscript and that you consider it suitable for publication in *Communications Biology*.

Sincerely,

Humberto G. Ferrón and Phil Donoghue

Evolutionary analysis of swimming speed in early vertebrates challenges the ‘New Head Hypothesis’

Humberto G. Ferrón^{1,2*} and Philip C. J. Donoghue^{1*}

¹School¹Palaeobiology Research Group, School of Earth Sciences, University of Bristol, Life Sciences Building, Tyndall Avenue, Bristol BS8 1TQ, UK. ~~e-mail:~~

humberto.ferron@bristol.ac.uk,
~~e-mail:~~

phil.donoghue@bristol.ac.uk

²Cavanilles Institute for Biodiversity and Evolutionary Biology, University of Valencia,

Paterna 46980, Valencia, Spain. E-mail: [humberto.ferron@uv.es](mailto:humberto.ferron@uv.es)

Abstract

The ecological context of early vertebrate evolution is envisaged as a long-term trend towards increasingly active food acquisition and enhanced locomotory capabilities culminating within the emergence of jawed lineagesvertebrates. However, support for this hypothesis has been anecdotal and drawn almost exclusively from the ecology of living taxa, despite knowledge of extinct phylogenetic intermediates that can inform our understanding of this formative episode. Here we analyse the evolution of swimming speed in early vertebrates based on caudal fin morphology using ancestral state reconstruction and evolutionary model fitting. We predict the lowest and highest ancestral swimming speeds ~~for~~in jawed vertebrates and microsquamous jawless groupsvertebrates, respectively, and find complex patterns of swimming speed evolution with no support for a trend towards more active lifestyles in the line leading to jawed groups. Our results challenge the hypothesis of an escalation of Palaeozoic marine ecosystems and shed light into the factors that determined the disparate

palaeobiogeographic patterns of microsquamous ~~and~~ versus macrosquamous armoured Palaeozoic jawless vertebrates. Ultimately, our results offer a new enriched perspective on the ecological context that underpinned the assembly of vertebrate and gnathostome body plans, supporting a more complex scenario characterized by diverse evolutionary locomotory capabilities reflecting their equally diverse ecologies.

MAIN TEXT

Introduction

The origin and early evolution of vertebrates is associated with a fundamental embryological revolution and recurrent rounds of whole genome duplication¹⁻⁴. In this context, the ‘New Head Hypothesis’⁵⁻¹⁰ proposes that the emergence of ~~the~~ neurogenic placodes, as well as neural crest and its novel derivative cell fates, brought about a remodelling of the head of invertebrate chordates and the development of novel sensory and anatomical structures, underpinning early vertebrate evolution. The ecological scenario underlying this evolutionary hypothesis is envisioned as a long-term trend from suspension feeding, in a *Branchiostoma*-like proto-vertebrate, towards increasingly active and predatory forms^{5-7,11}, culminating in the emergence of the first jawed groups (i.e., placoderms)^{12,13} and the subsequent diversification of the crown-gnathostome clades that dominate current vertebrate diversity (i.e., osteichthyans and chondrichthyans)^{14,15}. Most of the changes leading to the origin of jawed vertebrates are interpreted as adaptations for improved ventilation and locomotory capabilities¹⁶⁻¹⁹, acquired successively, as evidenced by the many phylogenetic and anatomical intermediate groups of ‘ostracoderms’ (i.e., jawless stem-gnathostome lineages)¹ (Fig. 1). However, ostracoderm biology is poorly constrained²⁰ and inferences of the ecological mode of ostracoderms, on which the New Head ecological scenario was constructed, are largely anecdotal¹⁰ or, at best, contested²¹⁻²⁹. This occurs because many

aspects of ostracoderm anatomy lack modern analogues. An exception to this is caudal fin architecture which has been shown to accurately predict swimming speed in living jawed fishes^{30,31} fishes with independence of other potential contributing factors³⁰⁻³². Here we extend this approach to living jawless fishes which have a distinct caudal fin structure from jawed fishes, like that of many ostracoderm stem-gnathostomes, demonstrating the same predictive value. Within this Extant Phylogenetic Bracket³² Bracket³³ we apply this approach to calculating swimming speeds in the ostracoderms, as a means of testing the New Head scenario of early vertebrate evolution as being characterized by an ecological trend toward increasingly active food acquisition⁵⁻¹⁰.

Fig. 1. Time-calibrated phylogenetic tree including all the Palaeozoic early vertebrate taxa considered in the present study. Taxa: Ve, Vertebrata; ~~Pe, Petromyzontidae~~; An, Anaspida; Pt, Pteraspidomorphi; Th, Thelodonti; Os, Osteostraci; Pl, Placodermi. Characters: a, unmineralized skeleton; b, mineralized dermal skeleton; c, ventro-lateral fins; d, perichondral bone ~~and~~, mineralized endoskeleton; ~~e~~, pectoral fins and girdles, epicercal tail and cellular bone; ~~f~~, jaws, ‘teeth’, pelvic fins, and girdles. Timescale: Lr, Lower; Md, Middle; Up, Upper; Llan, Llandovery; We, Wenlock; Lu, Ludlow; Pr, Pridoli; Mi, Mississippian; Pe, Pennsylvanian.

To achieve this, we present a new modelling approach for predicting swimming speeds in living fishes based on a number of caudal fin metrics, implemented using phylogenetic generalized linear models. Previous models are limited in their applicability as they are based in datasets with low taxonomic diversity and/or do not account for phylogeny^{30,31}. We use this framework to explore the phylogenetic and temporal patterns that characterize the

evolution of swimming speed in early vertebrate evolution, as a key component and predictor of their locomotory capabilities, by applying ancestral character state reconstruction and evolutionary model fitting.

Results

Building a model with living taxa to predict swimming speeds

We compiled 160 swimming speed records for 61 living fish species, including both osteichthyans and chondrichthyans with body lengths ranging from 4 cm to 105 cm and representing a wide range of ecologies, swimming modes and locomotion types (Supplementary Table 1). Using this dataset, we performed different phylogenetically informed regressions (PGLS) for predicting swimming speeds by checking combinations of multiple predictors (i.e., total body length, caudal fin shape metrics, swimming conditions and several locomotion parameters). Model selection analysis based on Akaike Information Criterion (AIC) shows that swimming speed is best explained by body length, swimming mode (burst, cruising), caudal fin height to width ratio, and swimming conditions (free, non-free-swimming) ($R^2=0.77$, Akaike weight=0.45) (Table 1 and Supplementary Table 1, Supplementary Fig. 1). Cross-validation analyses support that the best fitted model is robust and has a high predictive power ($R^2=0.73$) (Supplementary Table 1 and Supplementary Fig. 2). In addition, predictions performed on an independent dataset of living lampreys closely match records of swimming speeds reported in the literature ($R^2=0.97$) (Supplementary Fig. 3 and Supplementary Table 2). This validates the application of this model in fishes not only with epicercal tails, but also hypocercal tails (i.e., with caudal fins where the dorsal lobe is larger or smaller than the ventral one, respectively), such as in many (but not all) of ostracoderm groups³³groups³⁴.

Table 1. Fitting of phylogenetically informed regressions. The best model is shown in bold. wAIC, Akaike weight. Predictors: Length, total body length; Mode, swimming mode (burst, cruising); LocType, locomotion type (anguilliform, carangiform, median/paired fin propulsion, thunniform); AR, caudal fin aspect ratio; CircCF, caudal fin circularity; RoundCF, caudal fin roundness; SolCF, caudal fin solidity; HeWiCF, caudal fin height to width ratio; Cond, swimming conditions (free swimming, non-free swimming).

	R ²	AIC	ΔAIC	wAIC
Speed ~ Length	0.22	183.63	183.14	7.66E-41
Speed ~ Length + Mode	0.73	20.91	20.42	1.65E-05
Speed ~ Length + Mode + LocType	0.75	11.42	10.94	1.89E-03
Speed ~ Length + Mode + AR	0.75	6.06	5.57	2.77E-02
Speed ~ Length + Mode + CircCF	0.74	13.18	12.69	7.88E-04
Speed ~ Length + Mode + RoundCF	0.73	22.79	22.30	6.44E-06
Speed ~ Length + Mode + SolCF	0.75	10.89	10.41	2.47E-03
Speed ~ Length + Mode + HeWiCF	0.76	4.53	4.04	5.95E-02
Speed ~ Length + Mode + HeWiCF + SolCF + RoundCF + CircCF + AR	0.76	8.58	8.09	7.86E-03
Speed ~ Length + Mode + HeWiCF + AR	0.76	6.45	5.97	2.27E-02
Speed ~ Length + Mode + LocType + AR	0.76	11.38	10.89	1.94E-03
Speed ~ Length + Mode + LocType + CircCF	0.75	12.99	12.50	8.66E-04
Speed ~ Length + Mode + LocType + RoundCF	0.76	12.82	12.34	9.40E-04
Speed ~ Length + Mode + LocType + SolCF	0.75	13.31	12.82	7.36E-04
Speed ~ Length + Mode + LocType + HeWiCF	0.76	9.56	9.08	4.80E-03
Speed ~ Length + Mode + LocType + HeWiCF + SolCF + RoundCF + CircCF + AR	0.77	14.20	13.72	4.71E-04
Speed ~ Length + Mode + LocType + HeWiCF + AR	0.76	11.56	11.07	1.77E-03
Speed ~ Length + Mode + HeWiCF + Group	0.76	4.44	3.95	6.22E-02
Speed ~ Length + Mode + HeWiCF + Cond	0.77	0.49	0.00	4.49E-01
Speed ~ Length + Mode + HeWiCF + Group + Cond	0.77	1.51	1.02	2.69E-01
Speed ~ Length + Mode + LocType + HeWiCF + Group	0.76	8.13	7.64	9.85E-03
Speed ~ Length + Mode + LocType + HeWiCF + Cond	0.77	5.72	5.23	3.28E-02
Speed ~ Length + Mode + LocType + HeWiCF + Group + Cond	0.77	5.20	4.71	4.25E-02

Reconstructing ancestral swimming speeds in early vertebrates

From this PGLS model, we next inferred the cruising swimming speed of 41 early vertebrate taxa with well-known postcranial anatomy, including Palaeozoic cyclostomes (Myxinidae and Petromyzontidae), jawless stem gnathostomes (Conodonts, Anaspida, Pteraspidomorphi, Thelodonti and Osteostraci), and a representative sample of jawed stem gnathostomes (Placodermi) (Fig. 1 and Supplementary Table 3). Cruising swimming speed was considered here as it represents a good approximation of activity and metabolic level in living taxa³⁵⁻³⁹.

We performed ancestral character state reconstruction of cruising swimming speeds in a sample of 4500 trees accounting for both phylogenetic and temporal uncertainty (Fig. 2A), i.e., considering alternative phylogenetic hypotheses and fossil calibrations (Fig. 2A). Swimming speeds inferred for the ancestral nodes of the main clades varied considerably, with the highest values predicted for anaspids, ~~thelodonts, and petromyzontids~~ (means (mean \pm SD equal to ~~1.6465 \pm 0.06, 1.48 \pm 0.02 and 1.39 \pm 0.01~~ body lengths per second, BL/s, respectively) and the lowest predicted for osteostracans, placoderms, and pteraspidomorphs (mean \pm SD equal to ~~0.76~~1.17 \$\pm\$ 0.03 BL/s, 1.15 \$\pm\$ 0.03 BL/s and 1.13 \$\pm\$ 0.10 BL/s, respectively). Intermediate speed values are predicted for ~~pteraspidomorphs and osteostracans~~ (means \pm SD equal to ~~1.14 \pm 0.03 and thelodonts (1.0930 \pm 0.0607 BL/s, respectively)~~ which largely overlap with the cruising swimming speeds predicted for the ancestral node of all vertebrates (mean \pm SD equal to 1.2029 \$\pm\$ 0.0506 BL/s). In the following, we examined whether differences in body size impact on our estimates. When normalizing results by body length (i.e., considering all taxa being 0.1 meter in total body length) the pattern remains mostly approximately the same, with the only exception being the ancestral node of petromyzontids and osteostracans all vertebrates that ~~shows~~shows comparatively lower speeds ~~values~~ (means \pm SD equal to 1.6665 \$\pm\$ 0.03, 1.5545 \$\pm\$ 0.02, 1.43 \$\pm\$ 0.03, 1.41 \$\pm\$ 0.01, 1.5139 \$\pm\$

0.01, 1.51 ± 0.01 , 1.35 ± 0.01 , and 1.3038 ± 0.01 and 1.29 ± 0.0102 BL/s for anaspids, thelodonts, ~~vertebrates~~, pteraspidomorphs, osteostracans, placoderms, and ~~petromyzontids~~ vertebrates, respectively) (Fig. 2B).

Fig. 2. Ancestral cruising swimming speeds of Palaeozoic early vertebrates. Results derived from ancestral character state reconstruction of **(A)** non-size-normalized speeds, **(B)** size-normalized speeds, and **(C)** caudal fin morphology. The outcomes of each analysis are summarized both as density trees with mapped ancestral speeds (in **A** and **B**) and caudal fin morphologies (in **C**); and density plots showing the ancestral speeds predicted for the main clades (left and right panels, respectively). Density trees include a subsample of 100 trees randomly selected from the original pool, while density plots are based on the whole pool of trees. Black outlines in **(C)** represent the average of all the caudal fin morphologies inferred for each selected node. Swimming speeds are in body lengths per second (BL/s). Taxa: Ve, Vertebrata; ~~Pe, Petromyzontidae~~; An, Anaspida; Pt, Pteraspidomorphi; Th, Thelodonti; Os, Osteostraci; Pl, Placodermi.

We explored whether inferred ancestral morphologies would impact on ancestral estimates of swimming speed. For this, ancestral cruising swimming speeds of the main clades were also inferred by reconstructing the caudal fin morphology of their respective ancestral nodes, in 1000 trees randomly sampled from the original pool, using geometric morphometrics and posteriorly including those reconstructions into the established PGLS model (Fig. 2C). The results from this approach are remarkably similar to those derived from direct ancestral reconstruction of swimming speeds. Thus, anaspids ~~and thelodonts~~ show the highest ancestral cruising swimming speed values, followed by ~~vertebrates~~~~thelodonts~~, pteraspidomorphs, ~~vertebrates~~, osteostracans, ~~petromyzontids~~ and placoderms, in decreasing order of magnitude (means \pm SD equal to 1.422453 ± 0.016 , 1.392360 ± 0.006010 , 1.331 ± 0.005 , 1.324332 ± 0.006010 , 1.319317 ± 0.007 , 1.290 ± 0.004 , 1.298 ± 0.004 and 1.281 ± 0.006005 BL/s, respectively).

This general pattern is also manifest when ancestral reconstruction analysis is repeated in all the original source trees, representing the main previously proposed hypotheses on the interrelationships of these taxa and accounting for alternative topologies potentially not captured in our pool of trees (Supplementary Fig. 4).

Fitting models of evolution to identify potential trends in swimming speed capabilities

We fitted and compared two different evolutionary models to the cruising swimming speed datasets of early vertebrates using the pool of 4500 trees (i.e., one model including a directional drift or ‘trend’ component and the other one considering evolution under Brownian motion). When considering non-size-normalized cruising swimming speeds (Fig. 3A), rate of evolution (σ^2) ~~and trait mean (θ)~~ parameter estimates are very similar for both

models ($\sigma^2_{\text{BM}} = 0.01290116 \pm 0.00120010$; $\sigma^2_{\text{Drift}} = 0.01290115 \pm 0.0012, 00010$) while slightly lower mean parameter values are estimated for the Brownian motion model ($\theta_{\text{BM}} = 1.19582987 \pm 0.04470500$; $\theta_{\text{Drift}} = 1.20194493 \pm 0.07222191$). The drift parameter estimated for the drift model is, on average, negative but very close to zero ($-0.00020016 \pm 0.00120019$). AIC scores suggest that Brownian motion model fits the data better than the drift model for all considered phylogenetic trees (note in Fig. 3A that AIC_{BM} is always usually smaller than the $\text{AIC}_{\text{Drift}}$ and, when larger, the difference is less than two units; $\Delta\text{AIC} = -1.93027488 \pm 0.4023$). ~~However, when 4157). When~~ considering size-normalized cruising swimming speeds ~~(Fig. 3B), slightly higher evolutionary rate and lower, the pattern remains mostly the same, with a more marked difference on the~~ mean parameter values ~~are~~ estimated for ~~the Brownian motion each~~ model ($\sigma^2_{\text{BM}} = 0.002110027 \pm 0.000260003$; $\sigma^2_{\text{Drift}} = 0.002040027 \pm 0.00024, 0003$; $\theta_{\text{BM}} = 1.50853750 \pm 0.00930185$; $\theta_{\text{Drift}} = 1.56664817 \pm 0.0240$). ~~The 024; θ drift parameter estimated for the drift model is always negative and usually stronger ($-\theta_{\text{Drift}} = -0.00210012 \pm 0.0004$). In this case, despite AIC scores suggest that Brownian motion model fits the data better in most of the considered phylogenetic trees, a similar or slightly higher fit is also supported for the drift model in a number of instances (note in Fig. 3B that the difference between AIC_{BM} and $\text{AIC}_{\text{Drift}}$ is positive for several trees 0003; $\Delta\text{AIC} = -0.54026877 \pm 0.55581703$) (Fig. 3B).~~

Fig. 3. Evolutionary model fitting of cruising swimming speed in Palaeozoic early vertebrates. Parameter estimates (σ^2 , rate of evolution; θ , trait mean; and θ drift, drift of the trait mean) for Brownian motion and drift evolutionary models fitted to the datasets of (A)

non-size-normalized speeds and **(B)** size-normalized speeds, respectively. ΔAIC represents the difference between AIC_{BM} and AIC_{Drift} .

Discussion

Our methodological framework allows us to constrain for the first time the relationship between swimming speed, body size and caudal fin morphology in living non-tetrapod vertebrates accounting for their phylogenetic relationships (Table 1). The best fitting model shows a high predictive power of precisely inferring swimming speed, both when considering the original dataset and the leave-one-out cross-validation procedure (Figs. S1 and S2 and Tables S1), outperforming most of the previous approaches for predicting swimming speeds in fishes^{30,31,34,35,36,40}. As such, it lays the foundations for future studies aiming to assess locomotion aspects from morphological data in both living and extinct taxa and represents a valuable tool for interrogating long-standing discussions on the ecological ~~scenarios~~scenarios that ~~underlie~~attempt to explain early vertebrate evolution.

By applying this model to extinct Palaeozoic cyclostomes and stem gnathostomes, we reveal that the early evolution of vertebrates was characterized by diverging locomotory capabilities (Fig. 2). All our analyses predict that thelodonts and anaspids ancestrally had the highest swimming speed capabilities among all other major groups of stem-gnathostomes considered here, whereas the first jawed vertebrates (i.e., placoderms) had the lowest (or one of the lowest) swimming speeds. These results are supported by two alternative approaches (i.e., by direct ancestral reconstruction from the swimming speeds predicted in the tips, and by predicting swimming speeds in ancestrally reconstructed caudal fin morphologies) and remain unaltered when accounting for the potential effect of body size (Fig. 2), as bigger fishes tend to have lower relative swimming speeds than the smaller ~~fishes~~³⁶fishes⁴¹. When

comparing the fit of different evolutionary models to our data, those assuming no trend (i.e., Brownian motion) ~~appear to better~~ explain better the trait distribution ~~in most cases~~, given a pool of trees accounting for both phylogenetic and temporal uncertainty. ~~Only in few instances, a~~ model considering a subtle drift towards decreasing swimming speed has the lowest AIC values, ~~however in only a few instances. However,~~ BM is still equally supported given that most of the Δ AIC values remain far below two units (Fig. 3).

Our study shows that lineages of microsquamous taxa (i.e., thelodonts and anaspids) had higher swimming capabilities than those characterized by the presence of rigid bony carapaces (macrosquamous ostracoderms e.g., pteraspidomorphs, osteostracans, and placoderms) (Fig. 2 and Supplementary Fig. 4). This provides robust support for previous proposals of ~~their~~ more active lifestyles^{25,26} ~~and thelodonts and anaspids, with~~ higher potential for migration, dispersal and colonization of a wider range of habitats in the former^{26,37} habitats^{26,42}. Therefore, this finding offers a causal framework for understanding the contrasting palaeobioeogeographic patterns that characterize each of these groups, where heavily armoured ~~forms~~ lineages have distributions marked by a strong endemism limited to areas connected by shallow continental shelves, whereas microsquamous taxa had more cosmopolitan distributions resulting from the crossing of deep-water oceanic basins³⁸⁻⁴³ basins⁴³⁻⁴⁸. The prediction of high swimming capabilities as the ancestral condition for anaspids and thelodonts demonstrates that the rise of active nektonic ~~forms~~ vertebrates long-predated the ~~Devonian~~ Devonian⁴⁹, challenging the hypothesis of an escalation of Palaeozoic marine ~~ecosystems~~⁴⁴ ecosystems⁵⁰. In consequence, our results reinforce the view that the colonization of the pelagic realm by vertebrates might have follow more complex patterns than previously ~~believed~~⁴⁵ believed⁵¹. Ultimately, our results allow us to reject the ~~existence~~ hypothesis of a long-term evolutionary trend towards higher swimming capabilities

in stem-gnathostomes, a pattern expected from the prevailing view that early vertebrate evolution was characterized by a stepwise acquisition of increasingly active lifestyles and modes of food acquisition^{5-7,11,16-19}. In detail, however, our data reveal a more complex pattern in which the assembly of vertebrate and gnathostome body plans was characterised by diverse evolutionary trajectories offor locomotory capabilities, ecologies²⁵⁻²⁸, and feeding mechanisms^{21-24,29}.

Methods

Dataset of living taxa

Our dataset consists of 160 swimming speed records of 61 living osteichthyan and chondrichthyan species compiled from the literature (Supplementary Table 1). For each record, we collected information relative to the fish total body length, swimming conditions (free, non-free swimming with specimens under controlled experimental conditions), locomotion type (anguilliform, carangiform, thunniform, median/paired fin locomotion⁴⁶locomotion⁵²), swimming mode (cruising, burst), and different caudal fin morphological metrics, including the height to width ratio, circularity [$4\pi(\text{area}/\text{perimeter}^2)$], roundness [$4*\text{area}/(\pi*\text{major_axis}^2)$], solidity [$\text{area}/\text{convex area}$], and aspect ratio [$(\text{height}^2)/\text{area}$]. When only standard or fork length measurements were available, those were transformed to total body lengths by implementing species-specific equations (Supplementary Table 1). Caudal fin morphological variables were measured in ImageJ v 1.53b⁴⁷53b⁵³ by creating a macro (i.e., recording a series of commands using the command recorder) automating batch-processing of binary caudal fin outlines obtained from photographs of fishes in lateral aspect. Species-specific body length equations, photographs, and information regarding the locomotion type of each species were obtained from FishBase⁴⁸FishBase⁵⁴.

Phylogenetic generalized least squares models

Several phylogenetically informed regressions (PGLS) were performed for predicting swimming speeds by checking combinations of multiple predictors (i.e., total body length, caudal fin shape metrics, swimming conditions and several locomotion parameters). Akaike information criterion (AIC) was employed to compare the goodness of fit for the different models. PGLS analyses were implemented in the package ‘caper’⁴⁹ using R⁵⁰. Multicollinearity was checked by calculating variance-inflation factors (VIF) and generalized variance-inflation factors (GVIF) with the R package ‘car’ v.3.0.12⁵⁷ and considering a threshold of \$GVIF^{(1/(2 \cdot Df))} = 2\$ (equivalent to \$VIF = 4\$ )⁵⁸ (Supplementary Table 4). The phylogeny employed in the PGLS analyses was built from previously published trees^{51,52} by pruning taxa not included in our dataset using the R package ‘ape’⁵³. Species with multiple swimming speed records were included as polytomies that were resolved randomly with zero branch length. The predictive power of the best-fitting model was assessed by performing leave-one-out cross validation.

Our training dataset is ~~mostly constituted by~~ composed of living taxa ~~with, the majority of which have~~ symmetrical or epicercal caudal fins. Given that some pteraspnidomorphs, anaspids and thelodonts possess hypocercal caudal fins, we explicitly evaluated the potential of our PGLS model for predicting swimming speeds in living fishes with this condition. With this in mind, we compared previously published records of swimming speeds in living lampreys (Supplementary Table 2) with predictions derived from our model. The test sample included six records of three different species (*Entosphenus tridentatus*, *Petromyzon marinus* and *Lampetra fluviatilis*) and a range of body sizes comparable to that of the considered stem-gnathostome taxa.

Dataset of extinct taxa and supertree construction

A total of 41 early vertebrates, representing all taxa with well-known postcranial anatomy, were considered in our analyses. The dataset includes Palaeozoic cyclostomes (Myxinidae and Petromyzontidae), jawless stem gnathostomes (Conodonta, Anaspida, Pteraspidomorphi, Thelodonti and Osteostraci), and a representative sample of jawed stem gnathostomes (Placodermi) (Fig. 1 and Supplementary Table 3).

A pool of 4500 phylogenetic supertrees including these taxa (plus Galeaspida) was generated accounting for both phylogenetic and temporal uncertainty. For this, we obtained 45 different topologies using matrix representation with parsimony in the R package

‘phangorn’⁵⁴ phangorn’⁶² from 37 source topologies, accounting for the main hypotheses on the interrelationships of these taxa present in the literature (Supplementary Fig. 4). Each of the obtained trees was time calibrated 100 times using the R package ‘paleotree’⁵⁵ paleotree’⁶³ by randomizing the tip age of every species within the chronostratigraphic unit, at age or subperiod rank, where their first appearance occurs. A minimum age constraint was set in the ancestral nodes of the main clades of stem-gnathostomes considering their first appearance in the fossil record (Conodonta, Furongian⁶⁴; Anaspida, Llandovery²⁰; Pteraspidomorphi, Darriwilian-Sandbian^{65,66}; Thelodonti, Sandbian²⁶; Osteostraci, Aeronian⁶⁷; Placodermi, Telychian-Wenlock)⁶⁸.

Ancestral cruising swimming speeds estimations

The ancestral cruising swimming speeds of the main clades of early vertebrates were estimated following two alternative procedures. First, we performed ancestral character state reconstruction from cruising swimming speeds predicted for the tips. For this, caudal fin

morphological variables were measured in the 41 fossil taxa from reconstructions in the literature (Supplementary Table 3), following the same procedure than described above, and their cruising swimming speeds were inferred from the best fitted PGLS model. We considered free-swimming conditions and maximum body lengths reported in the literature (Supplementary Table 3). Ancestral character state reconstruction analysis was performed in the pool of 4500 phylogenetic trees using maximum likelihood method implemented in the R package ‘phytools’⁵⁶ phytools’⁶⁹. The same analysis was repeated with size-normalized cruising swimming speeds, considering all taxa having a total body length of 0.1 meters. Secondly, we reconstructed ancestral caudal fin morphologies and derived cruising swimming speed predictions from them. For this, we performed a geometric morphometric analysis on the caudal fin of the 41 early vertebrate taxa (excluding Galepasida where specimens with complete caudal fins are not known). We considered a total of 102 landmarks, including two landmark type I in the caudal fin base dorsal and ventral margins and 100 landmark type III equally interpolated along the caudal fin outline (Supplementary Fig. 5), that were digitized using TpsDig v.2.26⁵⁷26⁷⁰ on previously published reconstructions in lateral view (Supplementary Table 3). Landmark coordinates of all specimens were fitted by implementing generalised Procrustes superimposition in the R package ‘geomorph’⁵⁸ geomorph’⁷¹ to remove variation in rotational, scale and translational differences between specimens. Ancestral morphologies were reconstructed in 1000 trees randomly selected from the original pool of trees also using ‘geomorph’⁵⁸ geomorph’⁷¹. Caudal fin variables were then measured in the predicted morphologies and their cruising swimming speeds were inferred from the best fitted PGLS model. In these analyses, we considered free-swimming conditions and all taxa having a total body length of 0.1 meters. In all cases, results were visualized as density plots and density trees with mapped ancestral speeds using the R packages ‘ggplot2’⁵⁹, ‘ggridges’⁶⁰ ggplot2’⁷², ‘ggridges’⁷³ and

‘ggtree’⁶⁴ggtree’⁷⁴. We also visualized the average of all the ancestral caudal fin morphologies inferred for each node by obtaining Z projections in ImageJ v 1.~~53b~~⁴⁷53b⁵³.

Finally, we performed ancestral character state reconstruction from size-normalized cruising swimming speeds predicted for the tips in all 37 compiled source topologies using maximum likelihood method after time calibration in the R packages ‘paleotree’⁵⁵paleotree’⁶³ and ‘phytools’⁵⁶phytools’⁶⁹.

Fitting of evolutionary models

We explored the presence of evolutionary trends in the swimming capabilities of early vertebrates by fitting an evolutionary model including a drift or ‘trend’ component to our datasets of both non-size-normalized and size-normalized cruising swimming. A second model considering Brownian motion (i.e., where the trait evolves via a ‘random walk’) was also fitted to the data and AIC was employed to compare the goodness of fit for both models. We estimated the rate of evolution (σ^2) and trait mean (θ) by finding the maximum-likelihood parameter values for each model. The drift of the trait mean was also estimated for the drift model. The fit of evolutionary models and parameter estimations were carried out in the original pool of 4500 phylogenetic trees using the R package ‘geiger’⁶²geiger’⁷⁵. Results were presented as density plots with mean and standard deviation values using the R package ‘ggplot2’⁵⁹ggplot2’⁷².

References

1. Donoghue, P. C. & Keating, J. N. Early vertebrate evolution. *Palaeontology* **57**, 879–893 (2014).
2. Marlétaz, F. *et al.* Amphioxus functional genomics and the origins of vertebrate gene regulation. *Nature* **564**, 64–70 (2018).

3. Heimberg, A. M., Cowper-Sal, R., Sémon, M., Donoghue, P. C. & Peterson, K. J. microRNAs reveal the interrelationships of hagfish, lampreys, and gnathostomes and the nature of the ancestral vertebrate. *Proc. Natl. Acad. Sci. U.S.A.* **107**, 19379–19383 (2010).
4. Donoghue, P. C., Graham, A. & Kelsh, R. N. The origin and evolution of the neural crest. *Bioessays* **30**, 530–541 (2008).
5. Northcutt, R. G. & Gans, C. The genesis of neural crest and epidermal placodes: a reinterpretation of vertebrate origins. *Q. Rev. Biol.* **58**, 1–28 (1983).
6. Gans, C. & Northcutt, R. G. Neural crest and the origin of vertebrates: a new head. *Science* **220**, 268–273 (1983).
7. Gans, C. Stages in the origin of vertebrates: analysis by means of scenarios. *Biol. Rev.* **64**, 221–268 (1989).
8. Gans, C. Evolutionary origin of the vertebrate skull. *The skull* **2**, 1–35 (in *The Skull: Patterns of Structural and Systematic Diversity* vol. 2 1–35 (University of Chicago Press, 1993).
9. Northcutt, R. G. The origin of craniates: neural crest, neurogenic placodes, and homeobox genes. *Isr. J. Zool.* **42**, S273–S313 (1996).
10. Northcutt, R. G. The new head hypothesis revisited. *J. Exp. Zool. B Mol. Dev. Evol.* **304**, 274–297 (2005).
11. Denison, R. H. Feeding mechanisms of Agnatha and early gnathostomes. *Am. Zool.* **1**, 177–181 (1961).
12. Miles, R. S. VI.—Features of Placoderm Diversification and the Evolution of the Arthrodire Feeding Mechanism*, *Earth Environ. Sci. Trans. R. Soc. Edinb.* **68**, 123–170 (1969).
13. Carr, R. K. Placoderm diversity and evolution. *Bull. Mus. ~~natl. hist. nat., Sect. C~~Natl. Hist. Nat.*, **4C** **17**, 85–125 (1995).
14. Anderson, P. S., Friedman, M., Brazeau, M. D. & Rayfield, E. J. Initial radiation of jaws demonstrated stability despite faunal and environmental change. *Nature* **476**, 206–209 (2011).
15. Friedman, M. & Sallan, L. C. Five hundred million years of extinction and recovery: a Phanerozoic survey of large-scale diversity patterns in fishes. *Palaeontology* **55**, 707–742 (2012).
16. Mallatt, J. Early vertebrate evolution: pharyngeal structure and the origin of gnathostomes. *J. Zool.* **204**, 169–183 (1984).
17. Mallatt, J. O. N.—Feeding ecology of the earliest vertebrates. *Zool. J. ~~Linn.~~Linnean Soc.* **82**, 261–272 (1984).
18. Mallatt, J. Reconstructing the life cycle and the feeding of ancestral vertebrates. In *Evolutionary Biology of Primitive Fishes*, 59–68 (London, Plenum Press, 1985).

19. Mallatt, J. Ventilation and the origin of jawed vertebrates: a new mouth. *Zool. J. ~~Linnean~~ Soc.* **117**, 329–404 (1996).
20. Janvier, P. *Early vertebrates*. (Clarendon Press, ~~Oxford~~, 1996).
21. Purnell, M. A. Microwear on conodont elements and macrophagy in the first vertebrates. *Nature* **374**, 798–800 (1995).
22. Purnell, M. A. Feeding in conodonts and other early vertebrates. In *Fossils as living organisms* in *Palaeobiology II*, 401–404 (Blackwell ~~Science~~ Publishing, 2001).
23. Purnell, M. A. Scenarios, selection, and the ecology of early vertebrates. In *Major events in early vertebrate evolution*, 188–208 (Taylor & Francis, 2001).
24. Purnell, M. A. Feeding in extinct jawless heterostracan fishes and testing scenarios of early vertebrate evolution. *Proc. Royal Soc. B* **269**, 83–88 (2002).
25. Ferrón, H. G. & Botella, H. Squamation and ecology of thelodonts. *PloS one* **12**, e0172781 (2017).
26. Ferrón, H. G., Martínez-Pérez, C., Turner, S., Manzanares, E. & Botella, H. Patterns of ecological diversification in thelodonts. *Palaeontology* **61**, 303–315 (2018).
27. Ferrón, H. G. *et al.* Computational fluid dynamics suggests ecological diversification among stem-gnathostomes. *Curr. Biol.* **30**, 1–6 (2020).
28. Ferrón, H. G. *et al.* Functional assessment of morphological homoplasy in stem-gnathostomes. *Proc. Royal Soc. B* **288**, 20202719 (2021).
29. Miyashita, T., Gess, R. W., Tietjen, K. & Coates, M. I. Non-ammocoete larvae of Palaeozoic stem lampreys. *Nature* **591**, 408–412 (2021).
30. Sambilay Jr, V. C. Interrelationships between swimming speed, caudal fin aspect ratio and body length of fishes. *Fishbyte* **8**, 16–20 (1990).
31. Fisher, R. & Hogan, J. D. Morphological predictors of swimming speed: a case study of pre-settlement juvenile coral reef fishes. *J. Exp. Biol.* **210**, 2436–2443 (2007).
- 32. Giammona, F. F. Form and function of the caudal fin throughout the phylogeny of fishes. *Integr. Comp. Biol.* **61**, 550–572 (2021).
- 33. Witmer, L. M. & Thomason, J. J. The extant phylogenetic bracket and the importance of reconstructing soft tissues in fossils. In *Functional morphology in vertebrate paleontology*, 19–33 (Cambridge University Press, ~~Cambridge~~, 1995).

- 3334. Larouche, O., Zelditch, M. L. & Cloutier, R. A critical appraisal of appendage disparity and homology in fishes. *Fish. ~~Fishes~~ Fish.* **20**, 1138–1175 (2019).
- 35. Ferrón, H. G., Holgado, B., Liston, J. J., Martínez-Pérez, C. & Botella, H. Assessing metabolic constraints on the maximum body size of actinopterygians: locomotion energetics of *Leedsichthys problematicus* (Actinopterygii, Pachycormiformes). *Palaeontology* **61**, 775–783 (2018).
- 36. Ferrón, H. G. Regional endothermy as a trigger for gigantism in some extinct macropredatory sharks. *PLoS one* **12**, e0185185 (2017).
- 3437. Ferrón, H. G., Martínez-Pérez, C. & Botella, H. The evolution of gigantism in active marine predators. *Hist. Biol.* **30**, 712–716 (2017).
- 38. Ferrón, H. G. Evidence of endothermy in the extinct macropredatory osteichthyan *Xiphactinus audax* (Teleostei, Ichthyodectiformes). *J. Vertebr. Paleontol.* **39**, e1724123 (2019).
- 39. Watanabe, Y. Y., Goldman, K. J., Caselle, J. E., Chapman, D. D. & Papastamatiou, Y. P. Comparative analyses of animal-tracking data reveal ecological significance of endothermy in fishes. *Proc. Natl. Acad. Sci. U.S.A.* **112**, 6104–6109 (2015).
- 40. Jacoby, D. M., Siriwat, P., Freeman, R. & Carbone, C. Is the scaling of swim speed in sharks driven by metabolism? *Biol. Lett.* **11**, 20150781 (2015).
- ~~35. Ferron, H. G. Regional endothermy as a trigger for gigantism in some extinct macropredatory sharks. *PLoS one* **12**, e0185185 (2017).~~
- 3641. Videler, J. J. *Fish swimming*. ~~vol. 10~~ (Springer Science & Business Media, ~~Salisbury~~, 1993).
- 3742. Sallan, L., Friedman, M., Sansom, R. S., Bird, C. M. & Sansom, I. J. The nearshore cradle of early vertebrate diversification. *Science* **362**, 460–464 (2018).
- 3843. Blicek, A. & Turner, S. Global Ordovician vertebrate biogeography. *Palaeogeogr. Palaeoclimatol. Palaeoecol.* **195**, 37–54 (2003).
- 3944. Sansom, R. S. Endemicity and palaeobiogeography of the Osteostraci and Galeaspida: a test of scenarios of gnathostome evolution. *Palaeontology* **52**, 1257–1273 (2009).
- 4045. Young, G. C. Biogeography of Devonian vertebrates. *Alcheringa* **5**, 225–243 (1981).
- 4146. Zhao, W.-J. & Zhu, M. Siluro-Devonian vertebrate biostratigraphy and biogeography of China. *Palaeoworld* **19**, 4–26 (2010).
- 4247. Žigaitė, Ž. & Blicek, A. Palaeobiogeography of Early Palaeozoic vertebrates. *Geol. Soc. Lond. Mem.* **38**, 449–460 (2013).

- 4348. Smith, M. P., Donoghue, P. C. & Sansom, I. J. The spatial and temporal diversification of Early Palaeozoic vertebrates. *Geol. Soc. Spec. Publ.* **194**, 69–83 (2002).
- 4449. Sallan, L. C., Friedman, M., Sansom, R. S. & Sansom, I. J. The Silurian nekton revolution: Paleozoic jawless fishes exhibited modern form-dependent modes of habitat use. in *Geological Society of America Abstracts with Programs* vol. 47 632 (2015).
- 50. Whalen, C. D. & Briggs, D. E. The Palaeozoic colonization of the water column and the rise of global nekton. *Proc. Royal Soc. B* **285**, 20180883 (2018).
- 4551. Klug, C. *et al.* The Devonian nekton revolution. *Lethaia* **43**, 465–477 (2010).
- 4652. Webb, P. W. Hydrodynamics and energetics of fish propulsion. (~~Bulletin of the Fisheries Research~~Bull. Fish. Res. Board of Canada, Ottawa, Can. **190**, 1–158 (1975).
- 4753. Schneider, C. A., Rasband, W. S. & Eliceiri, K. W. NIH Image to ImageJ: 25 years of image analysis. ~~Nat.~~Nature methods **9**, 671–675 (2012).
- 4854. Froese, R. & Pauly, D. FishBase. www.fishbase.org (2020).
- 4955. Orme, D. *et al.* *The caper package: comparative analysis of phylogenetics and evolution in R*. ~~R package version 5~~ (2013).
- 5056. R Development Core Team. *R: A language and environment for statistical computing*. (R Foundation for Statistical Computing, ~~Vienna (2019)~~ (2020).
- 5157. Fox, J. & Weisberg, S. *An R Companion to Applied Regression*. (SAGE Publications, 2018).
- 58. James, G., Witten, D., Hastie, T. & Tibshirani, R. *An Introduction to Statistical Learning: with Applications in R*. (Springer, 2017).
- 59. Vélez-Zuazo, X. & Agnarsson, I. Shark tales: a molecular species-level phylogeny of sharks (Selachimorpha, Chondrichthyes). *Mol. Phylogenetics Phylogenet. Evol.* **58**, 207–217 (2011).
- 5260. Betancur-R, R. *et al.* The tree of life and a new classification of bony fishes. *PLoS Curr.* **5**, e1001550 (2013).
- 5361. Paradis, E. *et al.* Package '~~ape~~-ape': Analyses of phylogenetics and evolution, version ~~2~~ (2019).
- 5462. Schliep, K. P. phangorn: phylogenetic analysis in R. *Bioinformatics* **27**, 592–593 (2011).
- 5563. Bapst, D. W. paleotree: an R package for paleontological and phylogenetic analyses of evolution. *Methods Ecol. Evol.* **3**, 803–807 (2012).
- 5664. Murdock, D. J. E. The 'biomineralization toolkit' and the origin of animal skeletons. *Biol. Rev.* **95**, 1372–1392 (2020).

- 65. Gagnier, P. Y. Sacabambaspis janvieri, vertebre ordovicien de Bolivie. 2. Analyse phylogenetique. *Ann. Paleontol.* **79**, 119–166 (1993).
- 66. Gagnier, P. Y. Sacabambaspis janvieri, vertebre ordovicien de Bolivie. 1. Analyse morphologique. *Ann. Paleontol.* **79**, 19–69 (1993).
- 67. Tinn, O. & Märss, T. The earliest osteostracan *Kalanaspis delectabilis* gen. et sp. nov. from the mid-Aeronian (mid-Llandovery, lower Silurian) of Estonia. *J. Vertebr. Paleontol.* **38**, e1425212 (2018).
- 68. Young, G. C. Placoderms (armored fish): dominant vertebrates of the Devonian period. *Annu. Rev. Earth Planet. Sci.* **38**, 523–550 (2010).
- 69. Revell, L. J. phytools: an R package for phylogenetic comparative biology (and other things). *Methods Ecol. Evol.* **3**, 217–223 (2012).
- 5770. Rohlf, J. *tpsDig2*. ~~Version 2.26~~. (Stony Brook University, ~~New York~~, 2016).
- 5871. Adams, D. C., Collyer, M., Kaliontzopoulou, A. & Sherratt, E. *Geomorph: Software for geometric morphometric analyses*. ~~Version 3.1.0~~. (2019).
- 5972. Wickham, H. *ggplot2: elegant graphics for data analysis*. (Springer, ~~Houston~~, 2016).
- 6073. Wilke, C. O. *Ggridges: Ridgeline plots in 'ggplot2'*. ~~Version 0.5~~ in *'ggplot2'*. (2018).
- 6474. Yu, G., Smith, D. K., Zhu, H., Guan, Y. & Lam, T. T.-Y. ggtree: an R package for visualization and annotation of phylogenetic trees with their covariates and other associated data. *Methods Ecol. Evol.* **8**, 28–36 (2017).
- 6275. Harmon, L. J., Weir, J. T., Brock, C. D., Glor, R. E. & Challenger, W. GEIGER: investigating evolutionary radiations. *Bioinformatics* **24**, 129–131 (2008).

Acknowledgments: We thank two anonymous reviewers for providing thoughtful and valuable comments on the manuscript. H.G.F. is recipient of a European Commission grant H2020-MSCA-IF-2018-839636. P.C.J.D. was funded by Natural Environment Research Council (NERC) grant (NE/P013678/1), part of the Biosphere Evolution, Transitions and Resilience (BETR) programme, which is co-funded by the Natural Science Foundation of China (NSFC); and Leverhulme Trust Research Fellowship (RF-2022-167).

Author contributions: H.G.F. and P.C.J.D. conceived the project. H.G.F. analyzed the data and interpreted the results, and H.G.F. and P.C.J.D. wrote the manuscript.

Competing interests: Authors declare that they have no competing interests.

Data accessibility: Data and R code are available from the Figshare database:

<https://figshare.com/s/cefdb9a21f0804ef6a37> (link only for review).

REVIEWERS' COMMENTS:

Reviewer #1 (Remarks to the Author):

The authors have answered my main query regarding the taxon sampling, and appear to have addressed the other reviewer's comments as well. I have no further comments to make.

Reviewer #2 (Remarks to the Author):

The revision and replies to the reviewers' comments have been done very well and in great detail, critical amendments concerning the ancestral state approach were made and I have no more concerns about the publication of this manuscript in Communications Biology. Best regards, M. Buchwitz, 6th June 2022